

# Understanding Snow Bedform Formation by Adding Sintering to a Cellular Automata Model

Varun Sharma[1], Louise Braud[1], and Michael Lehning[1,2]

[1]School of Architecture, Civil and Environmental Engineering, Swiss Federal Institute of Technology, Lausanne, Switzerland
[2]WSL Institute for Snow and Avalanche Research SLF, Davos, Switzerland

**Correspondence:** Varun Sharma (varun.sharma@epfl.ch)

**Abstract.**

Cellular automata based modelling for simulating snow bedforms and snow deposition is introduced in this study. The well-known RESCAL model, previously used for sand bedforms, is adapted for this purpose by implementing a simple sintering model. The effect of sintering is first explored for solitary barchan dunes of different sizes and flow conditions. Three types of behaviour are observed: small barchans continue their motion without any perceptible difference while large barchans sinter immediately. Barchans of intermediate size split, leaving behind a sintered core and a smaller barchan is formed. It is found that sintering introduces an upper limit to the size of bedforms that can remain mobile. The concept of "maximum streamwise length" (M.S.L) is introduced and M.S.L is identified for different wind speeds using the solitary dune scenario. Simulations of the full evolution from an initially flat snow layer to a complex dune field are performed next. It is found that the largest bedforms lie below the M.S.L threshold. Additionally, it is found that shallow snow layers are the most susceptible to mechanical destabilization by the wind.

## 1 Introduction

Under the action of wind blowing over a layer of freshly deposited snow, snow re-organizes due to aeolian transport mechanisms into a number of shapes and bedforms; an initially flat surface may evolve into an undulated surface with significant height variations due to bedforms of various length scales. These bedforms and thus the surface continue to evolve until the wind speed falls below a threshold value. Thus upon snowfall, a snow grain lying on the surface may traverse a long and complex path before reaching its final resting place or in other words, until *ultimate deposition*.

Transport of snow by the wind and the formation of bedforms affects nearly all snowpacks, that by some estimates cover approximately 8% of the earth's surface during the course of an year (Filhol and Sturm, 2015). Research in aeolian transport of snow can be broadly divided into two streams based on their direct interaction with human activities. One stream deals with implications of snow transport on hydrology, particularly with respect to preferential deposition of snowfall (Lehning et al.,



2008; Gerber et al., 2018), sublimation of drifting and blowing snow (Déry and Yau, 2002; Sharma et al., 2018), avalanche prediction (Bartelt and Lehning, 2002; Schirmer et al., 2009) and road safety (Tabler, 2003). The other stream of research is focussed on polar regions where extensive snowpacks exist and are critical in modulating the energy and water budget of the Earth System (Vaughan et al., 2013). In almost all of such a vast array of topics, the implications of snow bedforms has not

be taken into account even though the physics of aeolian transport of snow and the morphodynamics of snow-covered surfaces are intimately linked.

The importance of snow bedforms for almost all snowpacks primarily stems from the effects of an undulating surface geometry on basic exchange parameters that dictate transfer of mass, energy and momentum between the surface and the atmosphere, namely the roughness lengths for specific humidity, sensible heat and velocity. Values of roughness lengths are

fairly unconstrained at the moment and severely affect modelling both for atmospheric flows (Amory et al., 2015; Vignon et al., 2017, 2018) as well as hydrology (Groot Zwaaftink et al., 2011) . Surface geometry and uncertainty in exchange processes across the snow-atmosphere boundary affects fields as distinct as interpretation of ice cores and palaeoclimatology (Birnbaum et al., 2010), remote sensing of snow-covered areas (Warren et al., 1998; Leroux and Fily, 1998; Corbett and Su, 2015; Picard et al., 2014) and both the mechanical and thermal dynamics of sea-ice packs (Petrich et al., 2012; Castellani et al., 2014).

Unlike research in snow bedforms, the study of bedforms in sand has progressed much further and can be considered to be at a fairly advanced state. There exists a vast body of literature documenting field, wind tunnel and numerical experiments of aeolian transport of sand and formation of surface morphological features ( see Kok et al. (2012) for a comprehensive review of these studies ) . Thus, almost all concepts of aeolian transport of granular material have been developed mostly in the context of the sand material. Fortunately, models as well as understanding developed in the sand context have been found to be readily

applicable in the snow context (Nishimura and Hunt, 2000; Comola and Lehning, 2017; Clifton et al., 2006; Doorschot and Lehning, 2002) . This is particularly true for *dry* and/or freshly fallen snow. This similarity extends to bedform features as well, with features such as waves, transverse dunes, barchans, longitudinal dunes which are found in both sand and snow surfaces.

One framework of understanding aeolian transport of any granular material, sand or snow, is to analyse grain-scale inter-actions between the grains themselves as well as grains and the air. When the stress at the surface due to wind motion ($\tau_s$)

exceeds a threshold value, the grains at the surface are dislodged and entrained into the air. This process is known as *aero-dynamic entrainment* (Bagnold, 1941; Anderson and Haff, 1991). Upon entrainment, a grain will follow a chaotic trajectory modulated by the turbulent air motions. Larger (and heavier) grains may fall back onto the surface with sufficient kinetic energy to dislodge additional grains into the air. This process is known as *splash entrainment* (Kok and Renno, 2009) . Particles impacting the surface may additionally rebound to re-enter the air through what is denoted as *rebound entrainment* (Anderson

and Haff, 1991). Parametrizations exist for each of the entrainment mechanisms and have been extended to sintered snow as well (Comola and Lehning, 2017; Schmidt, 1980)

At a larger scale however, grain-scale interactions alone are insufficient to explain the variation of fluxes of material as spatial heterogeneities of wind and surface shear stress begin to play an important role (Charru et al., 2013). Often, this variation is caused by an undulating topography at different scales. At any given instant in time, different locations on the surface, even

in close proximity, could be subjected to very different surface shear stresses (Groot Zwaaftink et al., 2014). In this scenario,



aeolian transport of snow and sand must be analysed in terms of regions dominated by net erosion or deposition. Regions of erosion are formed at locations with a increasing surface stresses in the horizontal direction ($\partial \tau_s / \partial x > 0$) whereas regions of deposition are formed where the shear stresses decrease ($\partial \tau_s / \partial x < 0$). Thus, there is a direct feedback between aeolian transport resulting in formation of bedforms, which in turn perturb the near-surface wind field.

Study of snow bedforms is distinguished by two prominent features of snow. First, and seemingly trivial, is the fact is that snow on the surface is replenished regularly (at least in the winter months in non-polar regions). This coupled with the fact that the time-scales of snow transport as much shorter than those of sand ( on account of the much lower density of snow compared to sand ) means that snow bedforms are ephemeral structures that form rapidly and then get buried during fresh snowfall.

The second, and more critical aspect from the perspective of surface morphodynamics is the ability of snow grains to form
ice bonds with each other in a process known as sintering. The process of sintering is quite complex and is dependent on the temperature, relative humidity and overlying pressure in the snowpack (Colbeck, 1997; Blackford, 2007; Gow and Ramseier, 1964). The effect of sintering on grain-scale aeolian processes is unknown at the moment. However some attempts to account for the effect of sintering in large-scale models has been reasonably successful. For example, regional-scale models used in Vionnet et al. (2014); Gallée et al. (2015); Amory et al. (2015); Agosta et al. (2018) use an erodibility factor as a decreasing
function of the age of snow. Thus at larger scales, the effect of sintering effect on aeolian transport can be considered in a straightforward manner; sintering prevents erosional activity activity leading to permanent deposition of snow.

It must be noted that moderately sintered snow bedforms can still be eroded by impacting snow grains during strong drifting and blowing snow conditions. This mechanism has been proposed to be the genesis of sastrugi (Leonard, 2009) which are thus classified as erosional bedform features. Sastrugi are one of the most common types of snow bedforms observed as opposed
to snow dunes such as barchans, waves etc. Their impact on near-surface wind flow is particularly pronounced as has been quantitatively described in recent studies focussed on Antarctica (Vignon et al., 2017; Amory et al., 2017, 2016) where it was found that flow perpendicular to the sastrugi field experiences two orders-of-magnitude higher drag.

Computational modelling has played a central role in improving the understanding of aeolian transport processes both at the grain scale as well as for developing parametrizations for mechanisms at different scales both in the context of sand and
snow transport. At the smallest relevant scale of grain-scale interactions, discrete element modelling (DEM) has allowed for linking material properties to transport mechanisms following the pioneering work of Anderson and Haff (1988). Studies using this technique have been particularly useful in understanding the saltation process (Carneiro et al., 2013, 2011; Pähtz et al., 2015). At a larger scale, particularly where (turbulent) air-grain interactions are also needed to be accounted for, the DEM technique or the grain-scale parametrizations described earlier are coupled with Reynolds-averaged Navier-Stokes (RANS)-
type fluid solvers where the full spectrum of turbulence is parametrized (Durán et al., 2014; Nemoto and Nishimura, 2004). In this regard, the recent use of large-eddy simulation (LES) technique to simulate turbulent flows is particularly promising (Sharma et al., 2018; Groot Zwaaftink et al., 2014; Dai and Huang, 2014).

From the perspective of resolving dynamics at the bedform-scale however, the above techniques are too computationally intensive. At this scale, there are two main modelling approaches. In one approach, the surface is treated as a continuum and




balance equations are derived for height at a point on the surface as a function of divergence of mass flux. (Anderson, 1987; Sauermann et al., 2001; Kroy et al., 2002) . The movement of mass through air is treated in an Eulerian fashion.

The alternative technique is to use cellular automata (CA) based models to simulate bedform dynamics. This technique is dramatically different from all previously stated methods that are essentially directly related to Newton's laws of mechanics and conversation laws. CA based models are purely of a phenomenological nature where mechanisms of erosion, transport and deposition are directly implemented in the form of *transitions* of state of cells containing the material of interest. Rules of transition for a cell are linked only to the state of the neighbouring cells and can be represented as time-dependent stochastic processes. These models are extremely attractive for their ability to seemingly reproduce features in complex systems in a quantitative and robust manner while being computationally parsimonious. The disadvantage is that due to a lack of firm footing in mechanics, the algorithms are not constrained and a lot of trial-and-error is involved in identifying relevant transition rules.

The genesis of CA type models is rooted in tenets of statistical mechanics, dynamical systems and chaos theory (Wolfram, 1984). Its application to bedform dynamics was pioneered by Werner (1995) who validated this approach by simulating formation and dynamics of various different dune shapes. CA models have been consistently improving over the past two decades with various works progressively updating the algorithms (i.e, the transition rules ) to approach better known and established physical laws. Notable examples include Nishimori et al. (1998); Bishop et al. (2002); Kocurek and Ewing (2005); Katsuki et al. (2005).

Narteau et al. (2009) introduced a new CA-based model named RESCAL that overcame a major shortcoming of the earlier CA based models by coupling the CA model of the granular material to a CA model for the air. This allowed for the first time to simulate the complete feedback between the evolving surface, the resultant perturbations in the flow and its impact on aeolian transport. Narteau et al. (2009) additionally provided a way of converting results of the CA-generated surface features into physically meaningful quantities that could be directly related to field data.

In this study we introduce CA based modelling for snow bedforms and snow deposition. Our work in this context is directly motivated by recent measurements of snow bedforms ( specifically barchans) made by our group in East Antarctica (Sommer et al., 2018). We adopt the RESCAL model as well as the methodology presented by Narteau et al. (2009) and further clarified by Zhang et al. (2012). We then implement a simple sintering model with the RESCAL model following the concepts described by Filhol and Sturm (2015). Description of the modelling framework is provided in Sect. 2. Upon establishing proper length and time scales for snow transport, we first simulate and describe the effects of sintering on solitary barchans in Sect. 3. Next, the full transition from an initially flat snow surface to a complex dune field and the effect of sintering on such a system is described in Sect. 4. Finally, in Sect. 5, the study is summarized along with an outlook for the role this methodology could play in the future.





## 2 The Cellular Automata approach

In this section we describe the cellular automata approach used in this study. Modelling the interaction between the wind, the snow-covered surface and the mobilized snow present in the air is achieved by running two cellular-automata type models in conjugation. One model is focussed on modelling the motion of the snow grains whereas the other focusses on the wind. As

noted in Sect. 1, we have adopted the open-source version of the RESCAL software that consists of implementation of both the models as well as their coupling. The RESCAL model is described in detail in Narteau et al. (2009) and its application for sand dunes is presented in Zhang et al. (2010, 2014) and thus we provide only a brief synopsis of the method for the sake of completeness. Full details of the models can be found in the aforementioned papers while the cohesion model is presented in this study.

### 2.1 Description of the CA model

Cellular automata modelling of snow transport consists of first discretizing the three-dimensional space to be simulated into cells that can only have one of three possible states: air, deposited snow (DS) or mobile snow. At the beginning of the simulation for example, one would typically consider a snow covered surface with no mobile snow cells. Each of these cells in this study have dimensions of $l_0$ in the horizontal directions and $h_0$ in the vertical direction. The ratio of the horizontal and vertical

length scales in a model parameter and in our case, $l_0/h_0 = 5$. This allows for greater vertical resolution in the modelling and is required for simulating snow bedforms that are typically much flatter and shallower than sand bedforms (Filhol and Sturm, 2015).

The model proceeds in the form of transitions of the cell states based on certain rules. In the RESCAL model, these rules are based on nearest-neighbour cell pairings know as doublets. A list of phenomenologically important doublet combinations is

made a-priori. Each of these doublet combinations can transition to a different doublet combination. These doublet groupings are presented in Fig. 1a, where they are grouped into six different phenomenological mechanisms, namely, deposition, erosion, transport, gravity, diffusion and avalanching. Before we elaborate further, it would be beneficial to avoid connecting the above listed mechanisms directly to physical processes with the similar names to avoid confusion. For example, erosion in the CA model is not directly linked to the different grain-scale entrainment mechanisms discussed in Sect. 1 even though at bulk-scale,

it is intended to produce the same effect.

Each of the transitions ( from doublet a to doublet b ) described in Fig. 1a are associated with a transition-rate parameter which has a dimension of inverse-time or frequency. This allows for a time-scale to be introduced into the CA model ( the length-scale being introduced by the discretization of the space ). The simulation progresses in a time-dependent stochastic fashion with each transition being regarded as a Poisson process. At each time step, all the doublets eligible for transition

are counted and a total transition rate of the system is computed, which is simply a total of all the transition-rates weighted by the number of doublets eligible for each transition (note that computing a global transition rate is possible for Poisson processes using the additive property). The global transition of the system is also a Poisson process and thus, the time-step can be stochastically chosen. The transition to be performed is chosen next based on the conditional probability of each transition.





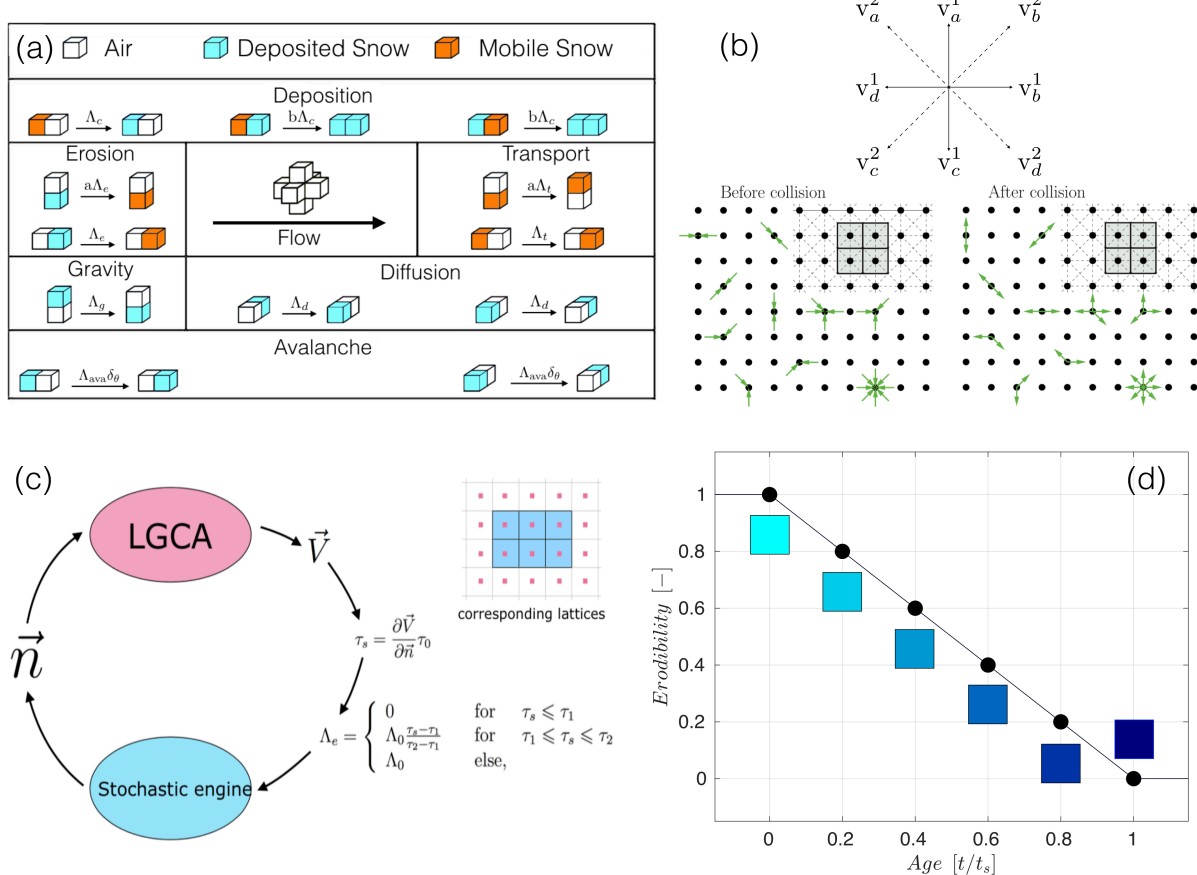

**Figure 1.** Description of the modelling framework. (a) Transition rules for CA of snow transport. (b) LGCA approach for simulating air flow. (c) Schematic of the coupling between the CA and the LGCA models. (d) A simple sintering model that reduces snow mobilization as a function of its age. Note that (a) and (b) are adapted from Narteau et al. (2009).

Finally, one of the doublets eligible for the chosen transition type is randomly picked and a transition is performed. In summary, at each instance of the simulation, the transition-type to be performed, the doublet to be transitioned as well as the value of the time-step itself are stochastically chosen based on the value of the transition-rates and the global state of the cells.

It is clear therefore, that only the relative values of the transition-rates are important. These rates are chosen to roughly reflect observations in reality. For example, at bulk-scale, transport is far more rapid than erosion or deposition. Erosion itself is typically faster but a more local process as compared to deposition which is slower but occurs over larger length scales (consider a scenario of a deposition of effluents from a plume). The processes of gravity and cross-stream diffusion (due to turbulence) are the fastest and slowest respectively. Avalanching is implemented in an extremely simple manner following Bak et al. (1988) - if the local slope is larger than a critical angle, a cell at that location is moved to a random location down the





slope. We realized a-posteriori that for the scale of the dunes simulated in this study, avalanching is of negligible importance and thus, it is not discussed further.

## 2.2 Description of the LGCA model

The flow overlying the evolving surface is simulated using the lattice-gas cellular automata (LGCA) approach. In this numerical
technique, the fluid is modelled as a set of particles lying on the nodes of a square (or cubical) mesh that is called a *lattice*. A particle must lie on one of the nodes of the lattice at all times and each node of the lattice can hold only one particle at any given time. Furthermore, a particle can move only to the nearest or next-nearest neighbours in one time-step of the simulation. In other words, the velocities that a particle may have are extremely limited, both in magnitude and direction. This is illustrated in Fig. 1b. A time-step in the LGCA consists of two sub-steps. The first is known as the *propagation* step
where all particles with non-zero velocities move to their destination lattice nodes. After this sub-step, it may happen that multiple particles (temporarily) lie on the same node. To impose the constraint that a node may have only one particle, multiple collisions between the incoming particles occur based on certain collision rules and the particle obtain new velocities. This step is known at the collision step. The situation after the propagation step (or before the collision step) is shown in Fig. 1b, left panel. Many nodes have incoming particles of differing number and directions. At each of these nodes, collision rules are
applied and the next propagation phase is shown in the right panel in Fig. 1b. As boundary conditions for the fluid particles, the collision of a particle with a solid object or a wall is modelled as a simple elastic collision ( similar to the model of the ideal gas! ). The LGCA methodology further provides a simple way to convert lattice-based velocities to "real" velocity of the flow. Typically it is simply the average of the velocities of particles in a given neighbourhood.

    The LGCA approach is in some sense a reduced-order model of the full Navier-Stokes equations achieved by imposing
strict constraints on directions and velocity values. Its development began with the pioneering work of Frisch et al. (1986) and was the precursor to more advanced Lattice-Boltzmann methods. The LGCA model in RESCAL is adopted from D'Humières et al. (1986). This modelling technique has the advantage that it is an extremely rapid method to simulate flow over complex surfaces, which is typically quite challenging for more traditional fluid simulation techniques such as large-eddy simulations or even Reynolds' averaged Navier Stokes (RANS) models. In the context of its use in this study, the LGCA, by simulating
flow over complex bedforms on the surface provides values of the surface-shear stress at every location of the surface. The surface-shear stress is essentially computed as a gradient of the velocity (computed using LGCA) in the direction normal to the local surface. The surface ( as well as the normal ) is of course the result of CA model for snow-transport.

    Of all the transition types, erosion is the only one directly linked to the morphology of the surface while all other transitions are independent of their location in 3-D space. This link between erosion and morphology is established by modifying the
transition rate for erosion according to location and making it a function of surface-shear stress, which, due to surface morphology is heterogeneous. Thus, areas in the domain with larger shear stress experience more erosion. In the RESCAL model, the erosion rate is linearly dependent on excess shear stress $(\tau_s - \tau_1)$ as,

$$\Lambda_e = \Lambda_0 \frac{\tau_s - \tau_1}{\tau_2 - \tau_1} \text{ for } \tau_1 \leq \tau_s \leq \tau_2, \tag{1}$$



where $\tau_s$ is the local surface shear stress and $\tau_1$ and $\tau_2$ are the lower and upper limits of the linear regime. For $\tau_s < \tau_1$, the erosion rate is set to zero while for $\tau_s > \tau_2$, the erosion rate is at the maximum possible values of $\Lambda_0$. The values of $\Lambda_0$ and $\tau_2 - \tau_1$ are constant and act only as a scale and slope parameters respectively. $\tau_1$ is equivalent to a threshold shear stress and is kept as a free parameter. As is explained in the following section, it is used to specify the wind speed. A list of transition rates

5   for the CA model used throughout this study are listed in Table 1.

| Model scales | Units | Value |
|---|---|---|
| $l_0$, length | cm | 32.5 |
| $h_0$, height | cm | 6.5 |
| $t_0$, time | See Table 2 | |
| Model Parameters | Units | Value |
| $\Lambda_0$, t.r for erosion | $1/t_0$ | 1 |
| $\Lambda_c$, t.r for deposition | $1/t_0$ | 0.5 |
| $\Lambda_t$, t.r for transport | $1/t_0$ | 1.5 |
| $\Lambda_g$, t.r for gravity | $1/t_0$ | $10^5$ |
| $\Lambda_d$, t.r for diffusion | $1/t_0$ | 0.005 |
| $a^a$, erosion coefficient | [–] | 0.1 |
| $b^b$, deposition coefficient | [–] | 10 |
| $\tau_2 - \tau_1$, linear regime for erosion | $\tau_0$ | 1000 |

[a]Ratio of vertical to horizontal rates for both erosion and transport mechanisms
[b]Enhancement factor for deposition due to DS type cells

**Table 1.** CA model scales and parameters using in this study. See Fig. 1a for more information about the transitions.

It is through the erosion mechanism that the flow simulation done using LGCA and the snow transport, done using CA is tightly coupled (see Fig. 1c). This robust coupling between the flow, its modification due to the undulating surface, and the evolution of the surface itself is the defining feature of the RESCAL model and thus makes it perhaps one of the best cellular automata models for simulating surface morphology that exist. This can be evidenced from the successful application of the

10   RESCAL model to simulate various complex sand bedforms on Earth (Zhang et al., 2010; Ping et al., 2014; Lü et al., 2017) and Mars (Zhang et al., 2012) and also gives us the confidence of introducing this model to the cryospheric science community.





In this study, we intend to focus on the effect of sintering on snow bedform dynamics. For this purpose we implement a simple sintering model that suits the CA modelling approach. Every time a cell transitions to the "deposited snow" (DS) type, the time-step of the simulation is noted. This provides a way of measuring the age of the snow cell, i.e, the period of time a snow parcel rests in one place. All transition rates for transitions consisting of DS type cells are then made to be a linearly decreasing function of the age of the cell with the most important transition being erosion. A sintering time, i.e, the time after which, a DS type cell cannot perform any further transitions must be chosen for such a model and we have chosen it to be 24 hours. Thus, after 24 hours, a DS type cell will permanently remain in the same location for the rest of the simulation. Cells that remain erodible, i.e., have ages less than 24 hours are henceforth denoted as eDS type cells while immobilized, sintered cells with ages greater than 24 hours are denoted as neDS type cells. This model is directly inspired by the approach of Filhol and Sturm (2015) and circumvents the requirement of accurate modelling of a highly complex and poorly understood phenomena of sintering. The erodibility of snow cells as a function of its age is shown in 1d. Note that in the context of this study, erosion and erodibility refer to the action of the wind. Erosion due to the impact of air-borne snow grains on a moderately sintered snow bedform are not simulated and are kept for future work.

## 2.3 Finding the length and time scales

A crucial contribution of Narteau et al. (2009), in addition to the development of the RESCAL model, was the development of a methodology of translating results of CA models, in which length ($l_0$) and time scales ($t_0$) are unknown, to real units, thus allowing for intercomparison between simulations and data collected from field experiments. We present their approach and the related calculations of length and time scales used in this study below.

Consider a system consisting of air blowing over a flat granular bed. If the flow velocity is faster than a threshold velocity, the system is mechanically unstable resulting in aeolian transport and leading to the formation of the bedforms. For a *small* period of time after aeolian transport commences, the system can be analysed using linear stability analysis which identifies the fastest growing mode of the evolving surface. Past theoretical analyses (Hersen et al., 2002; Elbelrhiti et al., 2005; Claudin and Andreotti, 2006) have established this length scale, $\lambda_{max}$ as being, equal to 50 $\rho_s \overline{d}/\rho_f$, where $\rho_s$ is the density of the grains, $\rho_f$ is the density of the overlying fluid ( air in our case) and $\overline{d}$ is the mean grain size diameter.

We performed a series of numerical experiments using the RESCAL model where we simulate flow over a wavy surface with a different wavelength for each experiment and with an amplitude ($A$) of $2\,h_0$, which is the smallest amplitude we can have in our discrete system. We allow the simulation to proceed for a very short time and measure the growth of the amplitude with time ($\mathrm{d}A/\mathrm{d}t$). The results of these numerical experiments is shown in Fig. 2a. It can be clearly seen that the fastest growing wavelength is $\lambda = 28l_0$. Thus,

$$\lambda_{max} = 28l_0 = 50\frac{\rho_s}{\rho_f}\overline{d}. \qquad (2)$$

Using values of $\rho_f = 1.00\,\mathrm{kg\,m^{-3}}$, $\rho_s = 910\,\mathrm{kg\,m^{-3}}$ and $\overline{d} = 200\,\mu\mathrm{m}$, we find the length scale of our model as $l_0 = 0.325\mathrm{m}$.

Once the length scale has been identified, we can now proceed to identify the time scale of the system. Returning to the system of air being blown over a flat surface, it has already been established that a *long* time after the initiation of the aeolian





transport and with the wind speed being constant, the flux of grains in the air achieves a steady-state value known as the saturated flux $Q_{sat}$. Past work, beginning already with Bagnold (1941) and refined over successive studies using both field and wind tunnel data has resulted in a semi-empirical formulation of $Q_{sat}$ as a function of material and flow properties (Bagnold, 1936; Iversen and Rasmussen, 1999; Ungar and Haff, 1987) . In effect, the saturated flux can be computed as

$$Q_{sat} = 25\sqrt{\frac{\overline{d}}{g}}\left(\frac{\rho_f}{\rho_s}\right)\left(u_*^2 - u_c^2\right) \text{ with} \tag{3a}$$

$$u_c = 0.1\sqrt{\frac{\rho_s g\overline{d}}{\rho_f}} \tag{3b}$$

where $u_*$ and $u_c$ are the friction velocity and the threshold velocity respectively.

If we consider an idealized scenario where the threshold velocity is zero with the resultant saturated flux value being $Q_{sat}^0$, a relationship relating saturated fluxes only as a function of $u_*$ and $u_c$ can be found as,

$$Q_{sat}^{ratio} = \frac{Q_{sat}}{Q_{sat}^0} = 1 - \frac{u_c^2}{u_*^2}, \tag{4}$$

This relationship is quite useful for CA based modelling as the free parameter of $\tau_1$ is essentially equivalent to $u_c^2$ and for modelling purposes can be chosen to be equal to zero. Thus, for different values of $\tau_1$, the CA model provides the L.H.S of (4). We performed a series of experiments beginning with the idealized scenario of setting $\tau_1 = 0$ and gradually increasing value of $\tau_1$ for each individual experiment. In each experiment we allowed the system to reach steady-state and calculated the steady-state flux of snow in the air. The resulting values of $Q_{sat}^{ratio}$ as a function of $\tau_1$ are shown in Fig. 2b. Once $Q_{sat}^{ratio}$ for different values of $\tau_1$ are found, we can compute the $u_*$ in real units for a given value of $\tau_1$. Note that $u_c$ is computed using Eq.(3b). Using the log-law, a given $u_*$ value can be converted to wind speed above a chosen height over the surface. Using a roughness length of $z_0 = 10^{-4}\,m$ , wind speed at a height of one meter above the surface, denoted as $U_{1m}$ is computed and shown in Fig. 2c. Once the $u_*$ value for each $\tau_1$ is found, we can compute the "real" saturated flux using (3a) and equate it to the "model" saturated flux. Thus $Q_{sat}^{model}l_0h_0/t_0 = Q_{sat}^{real}\text{m}^2/\text{s}$. As the length scales $l_0$ and $h_0$ are known, $t_0$ can be found as a function of $\tau_1$. Values of the time scale $t_0$ for different values of $\tau_1$ are shown in Fig. 2d.

Having established the length and time scales of the CA model, we choose three particular values of $\tau_1$ that are typically used in this study. These are $\tau_1 \in \{5, 20, 60\}$ and are equivalent to $U_{1m} \in \{20.5, 12.5, 7.0\}\,\text{ms}^{-1}$ and denoted further as high-wind (UH), medium-wind (UM) and low-wind (UL) cases respectively. Values of different quantities involved in calculating the time scales for these three $\tau_1$ values is provided in Table 2 with the full dataset of computed for all the values of $\tau_1$ provided in Supplementary Table 1.

## 3 Results I: Dynamics of fully-developed snow barchans

In the first set of results we consider the motion of a solitary barchan dune and the effect of sintering of its motion and morphology. All simulations in this section follow a common setup. The initial condition consists of a conical pile of unsintered snow with a given height and diameter of the base. We then choose three different wind speeds as noted in Table. 2 and allow





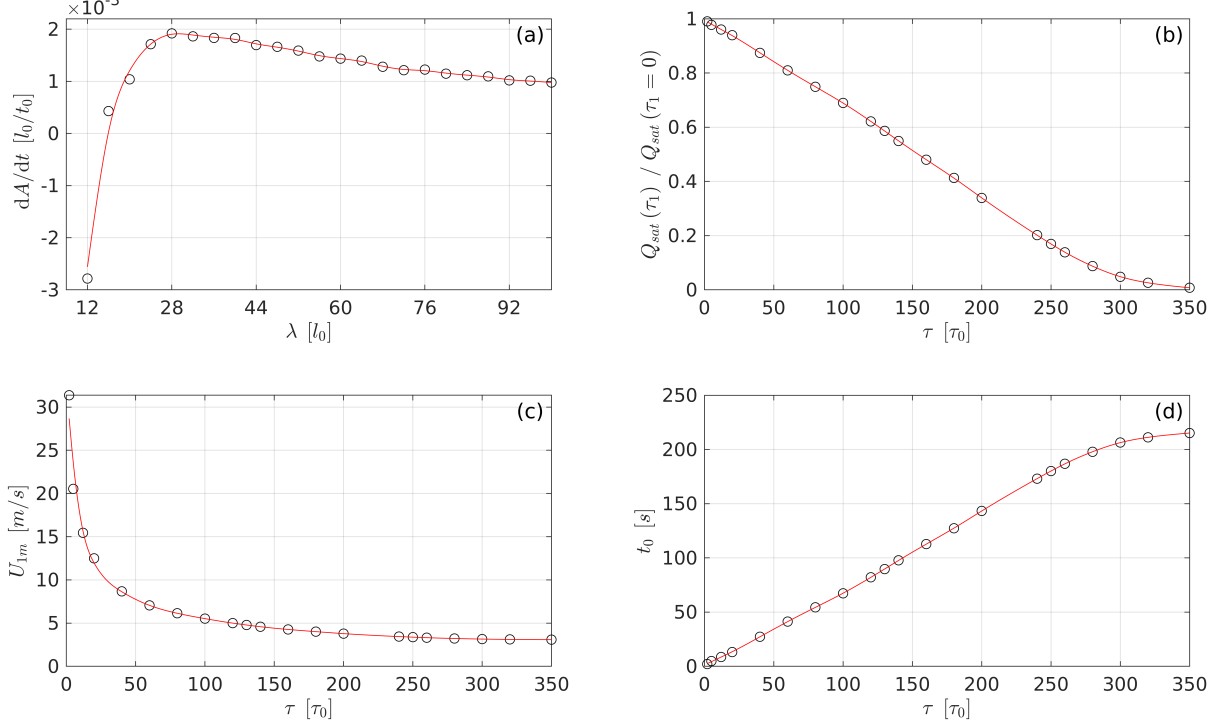

**Figure 2.** Establishing the length and timescales of the CA model. (a) Identifying the more unstable wavelength ( see Eq. 2 ). $\lambda_{max} = 28l_0$. (b) Variation of the $Q_{sat}^{ratio}$ with $\tau_1$. (c) Variation of the velocity at one meter above the surface $U_{1m}$ as a function of $\tau_1$. (d) Identifying the time-scale $t_0$ of the model for different values of $\tau_1$

the motion of snow particles. The effect of sintering is activated only once the dune is in steady-state motion. The lateral boundary conditions are periodic for both the flow as well as the particles with the particles' cross-stream location chosen randomly. Simulations are denoted as C$\alpha$, where $\alpha$ represents the height of the initial cone-pile in $h_0$ units. Relevant quantities for specific dune simulations are presented below while details for all the dunes simulated are presented in Supplementary

5   Table 2.

    Before discussing the results of these simulations, it is pertinent to place their purpose in the proper context. It is of course improbable that in reality, a conical pile of fresh snow would be found that is further moulded by the wind into barchans. The genesis of barchans, either in snow or even in sand is not well understood at the moment. The cone-pile experiments allow for creating realistic barchans without having to describe their genesis. Additionally, the size of cone provides some guidance

10  as to the dimensions of the barchan ultimately formed and thus allows for creation of barchans with a range of dimensions. Secondly, the effect of sintering in reality would begin as soon as snow is deposited on the surface, most likely through snowfall. By activating the effect of sintering on barchans in steady motion, we intend to isolate the interplay between the inertia of a barchan (which is a function of barchan size and wind speed ) and the effect of sintering which essentially acts as a damper for dune movement.





| $\tau_1$ | $Q_{sat}^{ratio}$ (model or real) [a] | $u_*$ [b] | $U_{1m}$ [c] | $Q_{sat}^{model}$ | $Q_{sat}^{real}$ [d] | $t_0$ [e] |
|---|---|---|---|---|---|---|
| $[\tau_0]$ | $[-]$ | [m/s] | [m/s] | $[l_0 h_0/t_0]$ | $[\mathrm{m^2/s}]$ | [s] |
| 5 | 0.9775 | 0.89 | 20.55 | $2.22 \times 10^{-2}$ | $9.643 \times 10^{-5}$ | 4.868 |
| 20 | 0.9394 | 0.54 | 12.49 | $2.14 \times 10^{-2}$ | $3.431 \times 10^{-5}$ | 13.148 |
| 60 | 0.8096 | 0.31 | 7.05 | $1.84 \times 10^{-2}$ | $9.42 \times 10^{-6}$ | 41.276 |

[a] computed using $\frac{Q_{sat}(\tau_1)}{Q_{sat}^0}$, where $Q_{sat}^0 = 2.273 \times 10^{-2} \, l_0 h_0/t_0$
[b] using Eq. 4, where $u_c = 0.134 \, \mathrm{ms^{-1}}$, using Eq. 3b
[c] through the log-law, $u = \frac{u_*}{\kappa} log\left(\frac{z}{z_0}\right)$, where $\kappa$ is the von Karman constant ( = 0.4) and $z_0$ is the roughness length $\left(= 10^{-4}\right)$
[d] computed using Eq. 3a using material properties as described in the text
[e] by equating model and real saturated flux

**Table 2.** Details of the chosen wind scenarios for further analyses along with calculation of different quantities leading up to finding the relevant timescales for each wind speed

## 3.1 Steady-state barchan motion

Under the influence of constantly blowing wind, a conical pile morphs into a barchan dune that starts to move downstream. In Fig. 3a, we show the top and side view of the evolution of the conical pile (case C20, with a constant wind speed of $U_{1m} = 20.5 \mathrm{ms^{-1}}$), into a barchan dune. Note that the time is scaled using the sintering time scale of 24 hours. The cone
essentially flattens and elongates in both the streamwise (along-wind) and crosswind direction, thus increasing the length and the width of the snow deposit. This evolution can be quantitatively seen in Fig. 3b. The quantities of length (L) and width (W) are shown on the left axis whereas, the height of the cone is shown on the right axis. The length and width of the dune initially is the same as the diameter of the cone, in this case, approximately 15 m while the height of the cone initially is 1.1 m. After approximately $3 \, t_s$, the morphology of the barchan, particularly its height are approximately constant and thus, we consider
that steady-state has been achieved. At steady state, the L,W and H dimensions of the C20 dune are respectively 26.4 m, 22.75 m and 0.59 m.

We additionally also show the evolution of the length of the longest streamwise section of the barchan. This quantity is termed as the maximum streamwise length or M.S.L of the bedform ($L_s$). It has been shown that this is the slowest moving part of a barchan and thus is representative of the speed of the dune (Zhang et al., 2014). At steady-state, we find $L_s$ to be
equal to 12.18 m. The relevance of $L_s$ will be more apparent in the coming paragraphs. It is interesting to note that it takes approximately $3 \, t_s$ or 72 hours for this dune to reach steady state. This is quite a long time considering that we have the wind blowing constantly for this period at $U_{1m} = 20.5 \mathrm{ms^{-1}}$. Thus morphodynamics of the barchans are much slower than typical sintering timescales. This point is elaborated upon further in the rest of the paper.





In Fig. 3c, we show four different measures of the *speed* of the dune. In the field, one would typically track the location of the crest of the dune as a function of time. Other measures are also possible, such as tracking the displacement of the horn or the tail of the dune. Location of the crest, tail and the horn of the barchan is tracked and plotted as a function of time. We find that all the speed measures are quite noisy with the horn-based measure being most noisy and thus one to be avoided. To find the speeds, we thus resort to finding the slope of the best-fit line, values of which are provided in the legend of Fig. 3c. The horn-based speed measure is found to be the fastest whereas the crest based measure is found to be the lowest. Admittedly, the differences are minor, especially compared to the fluctuations itself. We use an additional measure based on tracking the center-of-mass (COM) of the dune. In the field, this measure may be obtained by assuming constant snowpack density and a laser scanner. This measure may be preferred as a more physically-based and un-biased measure of dune movement and thus in the rest of this study dune speed is meant to be the speed of the COM of a dune.

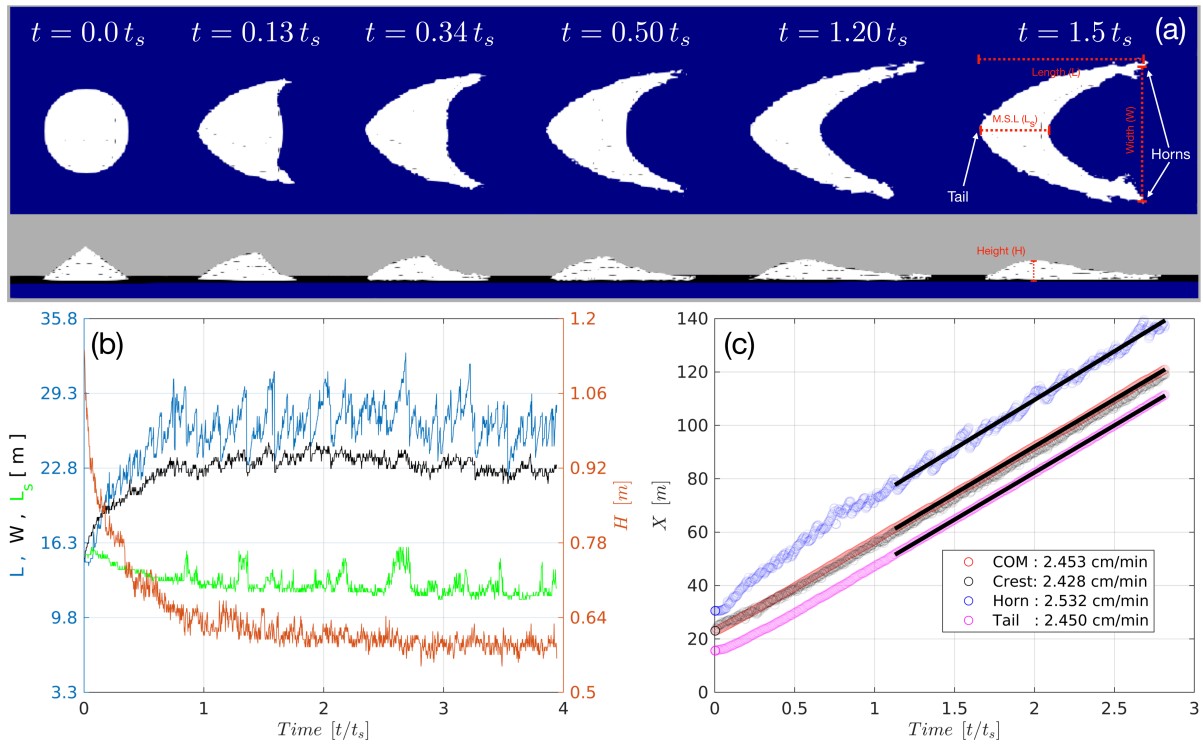

**Figure 3.** Morphodynamics of a solitary dune (case C20, high-wind scenario $U_{1m} = 20.5 \text{ms}^{-1}$) : (a) Visual representation of evolution from a cone-pile to a barchan. The final image is annotated for identifying different descriptors of a barchan.(b) Evolution of the length (L), width (W) and the maximum streamwise length ($L_s$) of the barchan. (c) Different versions of calculating the dune speed using the displacement of either the center-of-mass (COM), tail, horns or crest of the dune.

An example of the cone-based experiments presented in the previous paragraphs and shown in Fig. 3 are repeated for 16 different cone (and thus barchan) sizes and two additional wind speeds, namely, the medium wind UM $\left(U_{1m} = 12.5 \text{ms}^{-1}\right)$ and low wind UL $\left(U_{1m} = 7.0 \text{ms}^{-1}\right)$ scenarios. In Fig. 4a, we show the variation of dune speed with dune length for four





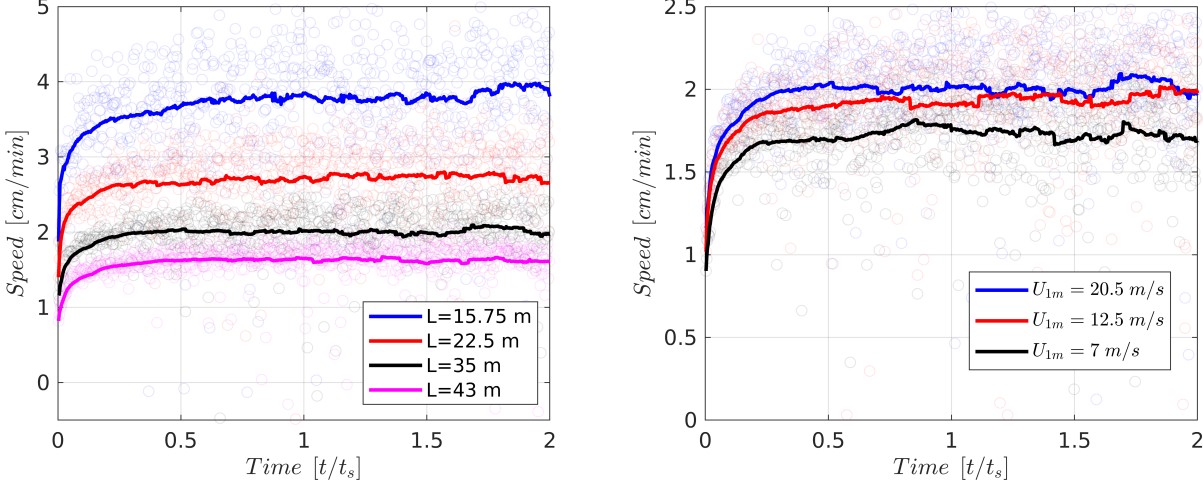

**Figure 4.** Influence of (a) dune height and (b) wind speed on dune velocity. The instantaneous speeds are represented by symbols while the time-averaged speed is shown by thick lines. The wind speed in (a) is the UH scenario whereas the dune in (b) is the cone C24

different dunes for the UH wind scenario. We show both the instantaneous dune speed ( computed every $50t_0$ steps, represented by circular symbols ) as well as the time-averaged velocity trends. This is to contrast the large instantaneous dune speed fluctuations with a comparatively constrained time-averaged value of dune speed. As expected, dune speed decreases with the size of the dune. In Fig. 4b we show the variation of the dune speed of the same dune for the three different wind scenarios.

Similar to the previous figure, there are large fluctuations of dune speed while the time-averaged value is more constrained. While differences between UH and UM cases are not significantly different, the dune in the UL case are almost 20 % slower than that in the UH and the UM cases. It must be noted that the effect of wind speed is not as significant as the dune size. Increasing the wind speed by 3 times between the UL and the UH cases does not seem to induce a proportional response from the dune speed.

Thus, we have shown that the dune speed is inversely proportional to its height and directly proportional to wind speed. This is in fact a well-know property of barchans first recognized by Bagnold (1941) and quantitatively explored using field-data by Elbelrhiti et al. (2005) for sand and by Kobayashi and Ishida (1979) for snow and in CA-type numerical experiments by Zhang et al. (2014). All these studies roughly proposed that

$$c^* = \frac{c}{Q} = \mathrm{f}\left(1/H\right) , \tag{5}$$

where $c$ is the dune speed, $Q$ is the saturated snow-flux, $c^*$ is the normalized dune speed, $H$ is the height of the dune and $f$ is a linear function. We explore this scaling for snow barchans in Fig. 5.

      Long time-averaging of both the height and velocity of the dune in each of the 48 simulations is done and the velocity-versus-height data points for all these simulations are shown in Fig. 5a. Simulations are classified based on wind speed alone and coloured accordingly. Note that velocity is normalized by the flux of snow in each simulation. We find that all the simulation

results, for different barchan dimensions as well as wind speeds lie on a hyperbolic function of H. The least-squares fit is found




to be

$$c^* = \frac{c}{Q} = \frac{a}{H+b} + d \,, \tag{6}$$

where $a$, $b$ and $d$ are parameters with values of 1.7 (dimensionless), -0.1 m and 0.94 m$^{-1}$ respectively. In the same figure, we also show the instantaneous values of normalized velocity as a function of height for each of the simulations. To quantify further the fluctuations in instantaneous velocity, we show histograms of this quantity for four different barchans for $U_{1m} = 20.5 \text{ms}^{-1}$ in Fig. 5b. We find that dune speeds become more constrained with increasing dune size with the smallest (largest) dune having the most (least) broad histogram.

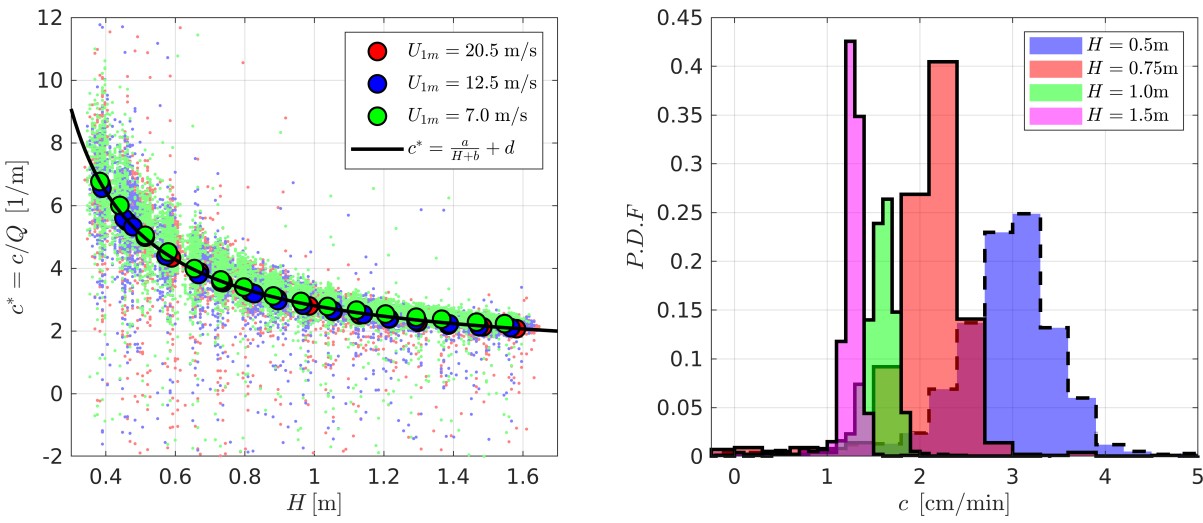

**Figure 5.** (a) Dune speed normalized by snow flux $(c/Q)$ as a function of height (H) for all the solitary dune simulations performed. Note that the instantaneous speeds are presented by small lightly-coloured symbols. Time-averaged dune speeds for each simulation are shown by large symbols. Note that all symbols coloured according to the wind speed of the simulation. (b) Probability distribution function of dune speeds for barchans of four different heights

We needed to perform time-averaging for at least 18 hours of instantaneous velocities to converge the time-averaged dune speed with the smallest dunes requiring up to 36 hours of averaging. While this is fairly simple to do in a idealized environment of numerical computations, it is highly unlikely that such time-averaging will be possible in the field primarily due to fluctuations of mean-wind speeds, effects of topography and the fact that the sintering process has similar timescales. On the other hand, due to large fluctuations of instantaneous velocity, a limited time-series, even of an order of a few hours is unlikely to show any systematic trend. A possible solution could be to sample multiple mobile dunes, hopefully of different sizes, at the same time using a laser scanner or photogrammetry.

We would like to note that values of time-averaged dune speeds found using the CA model are fairly similar to those reported in (a few) published studies ( see for example, Table 2 in Filhol and Sturm (2015) ). The values for the UH and the UM cases in particular are close to the studies by Doumani (1967), Kuznetsov (1960) and Kotlyakov (1961). It must be noted that these





studies are quite old and in future work, we shall further compare our results with the latest dataset from Kochanski (2018).
Dune dimensions and speeds for all the 48 simulations performed are provided for reference in Supplementary Table 2.

## 3.2 Effect of sintering on barchan motion

We now turn our attention to understanding the effect of sintering on dune morphodynamics. In this section, we focus on the

5 effect of sintering on barchans that are already in steady motion. This is achieved by "switching-on" the effect of sintering only
once the morphology and the mean dune speed has reached a steady state.

Sintering is activated in all 48 simulations with different barchan shapes and wind speeds. In general, three types of behaviours are observed. Firstly, the fastest-moving barchans continue their motion without much difference. On the other hand,
the slowest-moving barchans seem to sinter "in-place", i.e, the barchan immediately ceases to move, without any significant

change in morphology. The intermediate behaviour is found for a range of dune speeds in between the extreme cases where a
small part of the dune is deposited as a an non-erodible layer while the dune continues to move, albeit with slightly reduced
dimensions on account of loss of mass due to sintering.

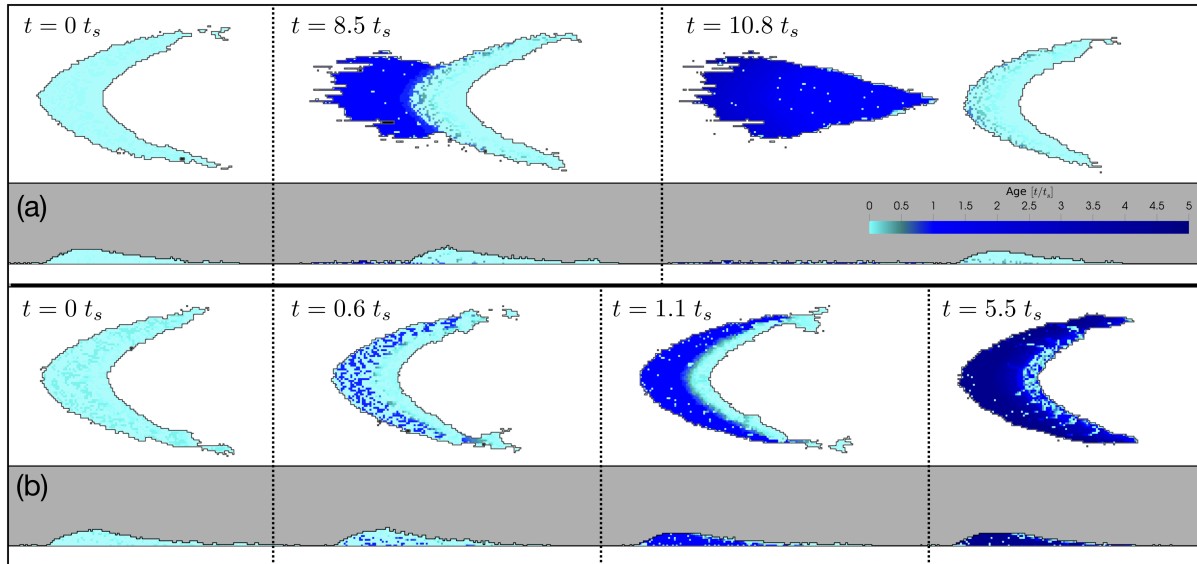

**Figure 6.** Effect of sintering on the morphodynamics of a mobile barchan (case C20) : (a) medium-wind $\left(U_{1m} = 12.5\mathrm{ms}^{-1}\right)$ and the (b) low-wind $\left(U_{1m} = 7.0\mathrm{ms}^{-1}\right)$ scenario. Note that the color scheme is such that light blue colors represent mobile (eDS type) cells while darker shades of blue are sintered(neDS type) cells.

To illustrate these three different types of behaviours, we show in Fig. 6, the morphodynamics of the same barchan dune
( C20 L,W,H,$L_s$ = 26.4 m, 22.75 m, 0.59 m and 12.3 m) for the UM and UL case. The UH case is not shown because no

perceptible difference in the morphodynamics is detected. For the UM case, shown in Fig. 6a, we find that the dune begins
to leave behind a mass of snow that can no longer be eroded. This mass originates at the tail-end of the dune which is the





oldest part of the dune as we shall see further. The dune velocity is such that there is a continuous ejection of mass from the tail as the dune continues its downwind motion. Ultimately, there is a split, where a sintered non-erodible mass of snow is left behind while a smaller barchan remain intact and mobile. In the UL case, the barchan sinters in place, and very quickly comes to a stand-still. Notice that the shape and the dimensions of the barchan in this case do not change much once the sintering is

activated. The morphodynamics of the two cases presented in Fig. 6 along two additional cases are provided as Supplementary Movies M1-M4.

The reason for this behaviour become clear when we analyse the distribution of ages of the constituent cells of a dune. Recall that "age" is defined as the time since a cell transitioned to the deposited snow (DS) type and has remained as such. In Fig. 7a, the three panels show the distribution of ages on the central slice of a C24 dune for three different wind speeds with

the UH and the UL cases consisting of the youngest and oldest barchan respectively. The age increases when moving from the leeward to the windward face of the barchan. The distribution in the flanks of the dune interior is also shown and it is found to be an extension of the distribution found in the central slice with no abrupt variations. It is interesting to note the stratigraphy of the dune and the distinct bands of each new layer added on the leeward face of the barchan. While the distribution indeed changes as a function of the wind speed, the banded pattern is consistent in all three panels. Furthermore, the banded pattern

also highlights the fact that deposition on the leeward side of the dune occurs in pulses rather than in a continuous fashion. This is correlated to the large fluctuations to dune speeds at short time scales as described earlier in Fig. 4 and 5.

The adjoining Fig. 7b quantifies the age distribution of the three panels in the form of cumulative distribution functions (C.D.F) of ages with respect to wind speeds. For the UH case, we find that all of the barchan is younger than $0.3t_s$ and thus, sintering does not have a perceptible impact on its dynamics. On the other hand, for the UM case, even though most of the

barchan is younger that the sintering timescale of 24 hours, approximately 50% of the barchan is older than $0.3t_s$. Thus, there is an increasing influence of sintering on the dynamics of this barchan, evidenced previously in Fig. 6a. Finally, in the UL case, almost 50% of the barchan has ages greater than the sintering time ! Thus when the effect of sintering is activated, the barchan almost immediately ceases its motion and comes to a halt thereby sintering "in-place".

Recall that for each wind speed (UL,UM and UH), we performed simulations for 16 different barchans with increasing

dimensions. These simulations allow us identify the size of the barchan at which the effect of sintering is strongly felt. To do so, in Fig. 8a, we show the location of the tail of the different barchans once the sintering is activated for the UH case. The slope of each of this lines would correspond to the tail-based speed measure shown previously in Fig. 3c. With increasing size, the barchan speed decreases as previously discussed. Lines are coloured green to indicate that the barchan speed does not change due to sintering. However, at a certain barchan size, there is a transition where, upon reaching sintering time $t_s$, the

tail no longer moves, i.e, the motion of barchan has been perturbed by sintering. We identify that for the UH case, this occurs for barchan with M.S.L, $L_s = 21.5$m ( line coloured black ), all barchans with sizes greater than this limit are also affected by sintering and show behaviour similar to that shown in Fig. 6b. A similar analysis for the UM case shows the limit to be at barchan size of $L_s = 12.3$m. In the UL case (not shown) all the chosen barchan shapes sinter "in-place" and thus the M.S.L is less than 8.8 m for this case.

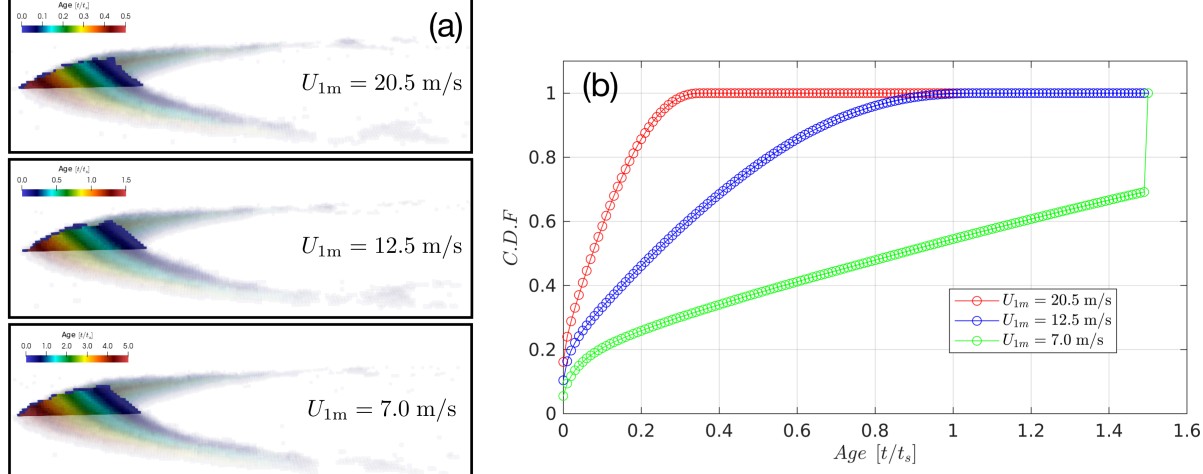

**Figure 7.** Distribution of age within a mobile barchan (case C24) prior to sintering: (a) Three panels show the distribution of age in the central slice of as well as the arms of the barchan for the three different wind scenarios. (b) Cumulative distribution function of the age of the entire barchan for the three different wind scenarios.

These numerical experiments highlight the fact that the interplay between barchan dimensions, wind speed and the sintering rate impose a maximum length scale of any snow bedform that can remain erodible and thus mobile. For example, in the UH case ($U_{1m} = 20.5 \mathrm{ms}^{-1}$), the largest streamwise dimension of any eDS-type bedform is limited to 21.5 m. Any bedform with dimensions greater than this limit (which perhaps arose earlier due to even higher wind speeds) will be strongly affected by

5 sintering and will most likely sinter in-place - thus converting to a neDS-type bedform. This limit decreases with wind speed and increases with the sintering rate.

In the following section, we move to a more realistic case where instead of beginning with a cone-pile or even a barchan in steady-motion, we directly simulate the effect of wind blowing over an initially flat fresh snow layer. As described earlier, the flat snow surface is mechanically unstable and rapidly evolves into various bedforms. Interestingly, we find that the results

presented in this section remain valid there as well.

## 4 Results II: From transverse waves to barchans

In the previous section, results of numerical simulations of solitary barchans were presented along with the influence of sintering on barchans in steady motion. This helped us identify the largest streamwise length scale that can exist in a mobile state with respect to wind speed and sintering rate.

We continue our investigation of the effect of sintering on snow bedforms in this section by showing results of simulations where air is blowing over an initially flat snow-covered surface. This is a more realistic scenario that can be considered to be equivalent to a scenario of strong winds after snowfall. It is also more realistic in the sense that we do not impose any particular



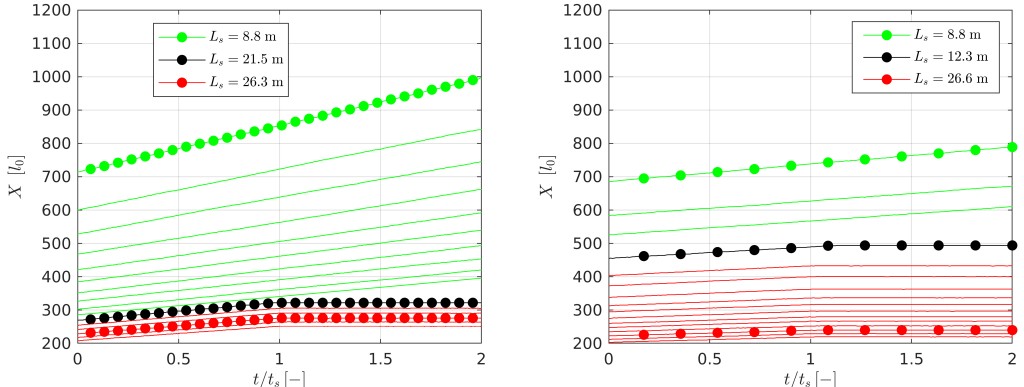

**Figure 8.** Identification of the maximum streamwise length (M.S.L, $L_s$) for (a) high-wind and (b) medium-wind scenarios. In each figure, the location of the tail of the barchan is plotted as function of time. Lines from top to bottom represent barchans with increasing size. The lines are coloured to identify mobile (green) and immobilised (red) behaviours. The black line identifies the barchan at which this transition occurs.

mobile bedform ( such as a barchan in the previous section ); bedforms such as merged barchans, transverse dunes, snow waves and sintered immobile snow deposits emerge through self-organization of snow. Finally, we activate the effect of sintering from the first time-step itself, once again reflecting our purpose to move towards simulating more realistic scenarios.

Transition of a flat granular bed to an undulating surface with various bedforms under the action of overlying fluid flow
has been investigated in the past, mainly in the context of the air-sand (aeolian) or water-sand (riverine) systems. Many such studies have in fact employed a CA-based framework similar to ours. There are two important mechanisms that these studies have revealed that are relevant for the present study. Firstly, it is now understood that there is a strong link between transverse surface waves and barchans which transition from one type to another as a function of sediment supply (Nishimori et al., 1998). Lack of sediment supply causes transverse waves to break-up and form barchans; on the other hand, providing additional
sediments causes barchans to link up in the cross-stream direction to form transverse waves again. Transverse waves can also break up due to topographical effects and due to suppression of cross-stream turbulent diffusion of grains, once again resulting in barchans. Secondly, it has been proposed that the tendency of any mobile bedform is to continuously increase in size in the presence of adequate wind speed and sediment supply, with the ultimate limit imposed by the size of the atmospheric boundary layer for aeolian system (Andreotti et al., 2009) (for riverine systems, the limit would be the depth of the river).

Our hypothesis is that the sintering process, over time, causes the erodible snow deposits to convert to becoming non-erodible, thereby depleting snow supply and increasing the occurrence of barchans as opposed to an equivalent system without sintering. Additionally, it would be interesting to check whether the maximum streamwise length of a mobile snow bedform found in the previous section is indeed found in this, more complicated system as well.

To confirm these hypotheses we perform simulations of flow over an initially flat snow layer of depth varying as 6.5cm,
32.5cm, 0.65m,1.3m. These simulations are denoted as H1,5,10,20 respectively, denoting the thickness in the CA height scale





$h_0$. This set of four simulations was performed for two different wind speeds, namely the UH and UM cases as in the previous

section. The entire set of simulations was repeated by deactivating sintering to provide a contrast and highlight the effect of

sintering. Thus, in total 16 simulations were performed.

5     Each simulation has domain size of 1000 $l_0$ x 1000 $l_0$ x 100 $h_0$ in CA units or equivalently, a domain of approximately 325 m

in the horizontal directions and 6.5 m in the vertical. Care was taken to ensure that the vertical extent of the domain is adequate

- additional simulations performed with larger heights showed no major differences. The horizontal boundary conditions in the

lateral directions were periodic.

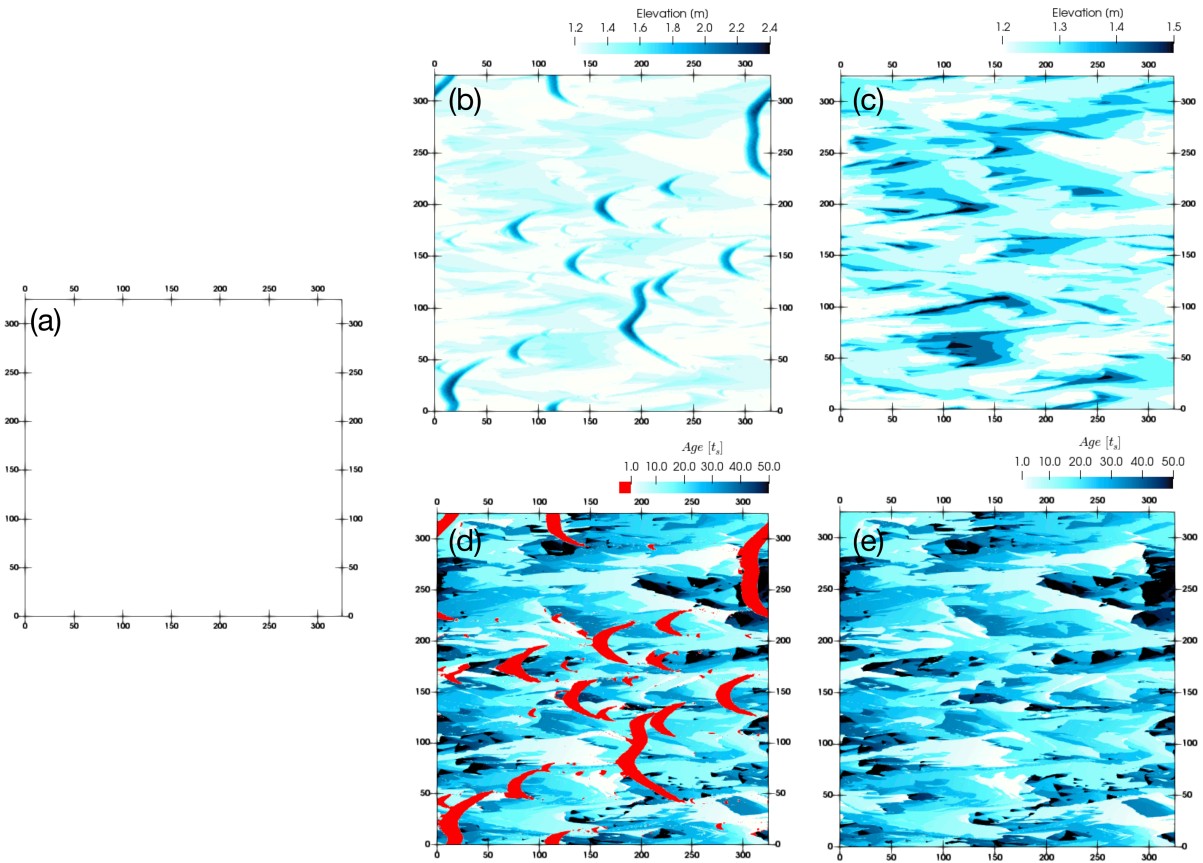

**Figure 9.** Evolution from an initially flat snow layer of depth 1.3 meters to a complex dune field. (a) Initial condition of the simulation. (b) Elevation map of the surface at $t = 50t_s$. (c) Elevation map of the sintered snow surface with the eDS type cells filtered out. (d) Age of the surface with eDS type cells colored in red. (e) Age of the surface with eDS type cells filtered out.

    An illustration of the evolution of such a simulation ( with sintering ) and the information that can be extracted is presented in

Fig. 9. Top-view elevation of the surface in the UH-H20 case ( i.e., $U_{1m} = 20.5\text{ms}^{-1}$, initial snow depth of 1.3 m ) is presented

10    at $t = 0t_s$ (subfigure a) and at $t = 50t_s$ (subfigure b). The initially flat surface is now re-organized into an undulating surface

with multiple barchans, some laterally merged barchans and also some large-scale snow deposits that are sintered (neDS-type



bedforms). The surface shown in Fig 9b is filtered to remove all eDS-type cells revealing the underlying non-erodible snow layers in Fig. 9c. Such sintered deposits cover most of the surface area with the difference between the highest and lowest points of the sintered mass being approximately 40 cm. The age of the surface is presented in Fig. 9d. The eDS-type cells are specially coloured red to highlight the fact that mobile bedforms are precisely the high elevation regions in subfigure b. Finally,

5  in Fig. 9e, the mobile snow cells are filtered out to show the distribution of age on the surface of the sintered mass. It is quite interesting to note the large distribution of ages with many clusters of old and new deposits in close proximity. Supplementary Movies M5-7 shows the full evolution from the flat surface to undulating surface consisting of barchans and sintered snow deposits.

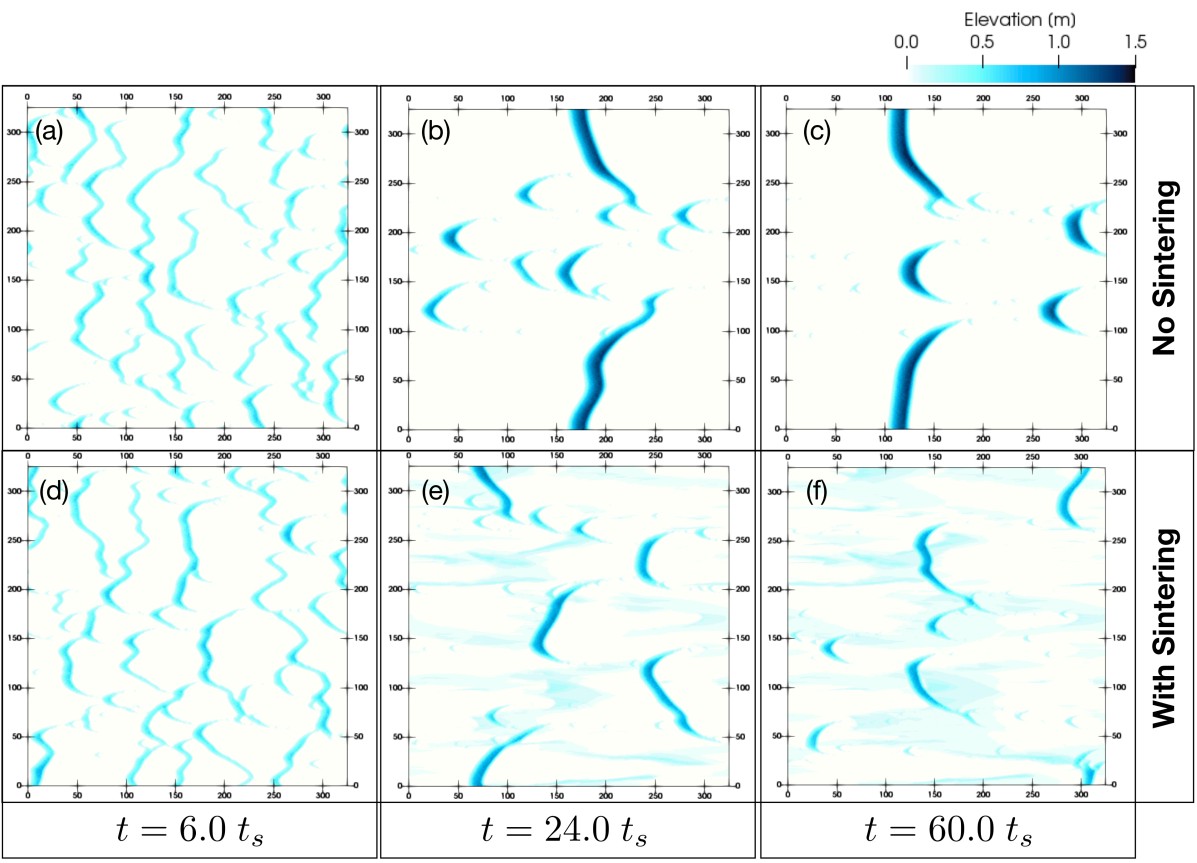

**Figure 10.** Evolution of an initially flat snow layer of depth 6.5 centimeters in the high wind (UH) scenario

The following two figures provide some more insights into the effect of sintering on the snow bedforms. In Fig. 10, all six

10  snapshots of the surface elevation maps come from the UH-H1 type simulation with the shallowest snow layer of depth 6.5 cm. The upper panels (subfigures a-c) shows results from simulations without sintering while the simulation results shown in the lower panels (subfigures e-f) account for the sintering effect. For comparison, the snapshots for the two simulations are presented at the three different times. In the left column panels (a,d), there is no major qualitative difference between the



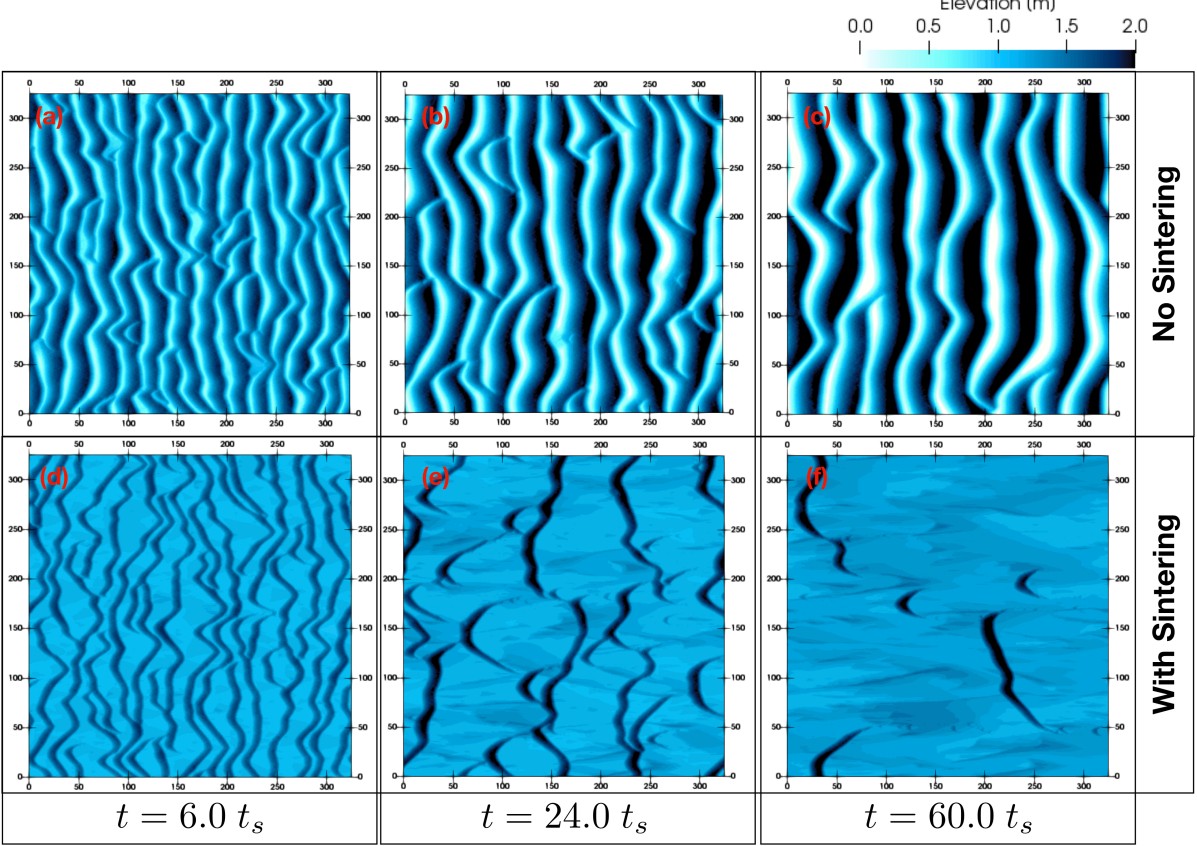

**Figure 11.** Evolution of an initially flat snow layer of depth 1.3 meters in the high wind (UH) scenario

simulations. Most bedforms have their dominant dimension in the cross-stream (transverse) direction and are quite limited in their streamwise extent. Moving to the middle column (b-e), difference between the two simulations begin to emerge. In the non-sintering case (panel b), we find that the eDS type cells are accumulated in a few bedforms consisting mainly of barchans and a long transverse dune. In comparison, in the with-sintering case (panel e), a few small barchans are found along with a

5 few transverse dunes. There are also a few neDS type cells forming large patches throughout the domain. Note also the fact that the dunes in panel b are higher than in panel e. In the final, right column, we find that in the non-sintering case, there are now a few barchans that have grown in size while the transverse dune is still present. In the corresponding with-sintering case (panel f), the neDS cell patches have grown in size and there are few eDS type bedforms, consisting of a few short transverse dunes and small barchans. The difference in the height of between non-sintering and with-sintering type simulations is even

10 more clear in the right most column.

With a initial snow-depth of 6.5 cm in the H1-type simulations, there is a deficit in the sediment supply needed to form large transverse waves and the initial transverse waves breakup into barchans. In the simulation without sintering, the individual barchans then grow in size. This was discussed at the beginning of this section and has been shown in previous studies focussed





in sand. Due to the additional sintering mechanism present in this study ( and indeed in snow in reality), there is an additional curtailment of sediment available for aeolian transport and for forming bedforms. Thus the bedforms in the case with sintering are smaller, flatter and more dispersed.

To further clarify the effect of sintering on snow bedforms, we additionally remove the constraint of sediment supply by
performing simulations of flow over a much deeper snow layer. These simulations denoted as UH-H20 have a uniform initial snow depth of 1.3 m, twenty times larger than the UH-H1 case presented previously. Results for the UH-H20 case are presented in Fig. 11 in a fashion similar to Fig. 10. Focussing first on the difference between the upper panels in Figs. 10 and 11, note that due to adequate supply of sediments, regular snow-waves are formed in the UH-H20 case as opposed to the UH-H1 case. In the case of UH-H20, the waves in fact seem to grow both in their streamwise length ( or wavelength) as well as height (
or amplitude). Results in the with-sintering case ( lower panels, subfigures d-f) are starkly different than the corresponding non-sintering case. Even at a comparatively early stage of the simulation, the snow bedforms in the with-sintering cases are much shorter in streamwise thickness along with shorter heights in comparison to non-sintering case. As time progresses, more and more eDS type cells are converted to neDS cells resulting in breaking up of the transverse bedforms and patches of sintered snow, similar to the with-sintering case in Fig. 10. At a later stage (panel f), most of the earlier bedforms have disappeared
completely, resulting in a few isolated barchans and short transverse dunes. In comparison to Fig. 11c, which shows large snow-waves, the results in panel f are more similar to Fig. 10f instead ( which it must be recalled had 20 times fewer snow sediments to begin with). Sintering indeed has a large impact on the bedforms that form on snow layers !

We concluded Sect. 3 with stating that the sintering mechanism imposes a maximum length scale (M.S.L) that a bedform can have to remain mobile. This length scale depends directly on wind speed and the sintering timescale. We restricted our
analysis to a single sintering rate of $t_s$ of 24 hours and only two windspeeds - UH ( $U_{1m} = 20.5\mathrm{ms}^{-1}$) and the UM case ($U_{1m} = 12.5\mathrm{ms}^{-1}$) which provided the maximum length values of $L_s = 21.5\mathrm{m}$ and $L_s = 12.3\mathrm{m}$ respectively.

In Fig. 12, we present the maximum streamwise length of snow bedforms present in the domain as a function of time for all simulations UH-H$\alpha$ (blue lines with symbols) and UM-H$\alpha$ ( red lines with symbols ) where $\alpha \in \{1, 5, 10, 20\}$. We additional mark the limits suggested in the analysis in Sect. 3 for the UH ($L_s = 21.5\mathrm{m}$, solid blue line) and the UM ($L_s = 12.3\mathrm{m}$, solid
red line) cases. All simulations have maximum length scale that ultimately line below the limits suggested in Sect. 3. Thus the concept of sintering limiting the largest mobile snow-bedforms, first developed from the solitary dune experiments seems to be quantitatively applicable even in a more complex (as well as realistic) scenario of surface evolution from an initially flat surface. Figure 12 provides additional information. In particular, it is found that for all the UH-H$\alpha$ cases, the M.S.L initially increases upto about 20 $t_s$ after which it begins to decrease and ultimately falls below the M.S.L limit at approximately 140
$t_s$. For the UM case, the initial increase in the M.S.L is found for only the H1 case, whereas for all the deeper snow layer simulations, the M.S.L decreases rapidly after $1t_s$ already. In the UM case it is also interesting to note that the M.S.L values remain constant after approximately 120 $t_s$ and that the largest M.S.L is found for H1 case. Indeed, even in the UH case, the H1 simulation shows slightly larger values of M.S.L as compared to the deeper snow pack simulations.

In the final figure of our analysis, Fig. 13, the number of eDS type cells in a simulation as a function of time is shown for
each of the UH-H$\alpha$ and UM-H$\alpha$ cases. As the initial condition for each of these simulations, recall that we consider a flat snow





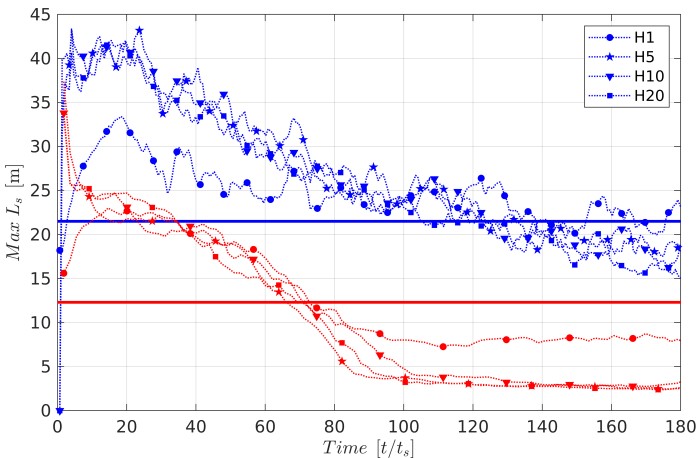

**Figure 12.** Identification of M.S.L for the different $H\alpha$ cases. Lines are colored to represent the high-wind(UH, blue) and medium-wind (UM, red) scenarios. Corresponding values of M.S.L identified in Sect. 3 and in Fig. 8 are shown as thick solid lines for reference.

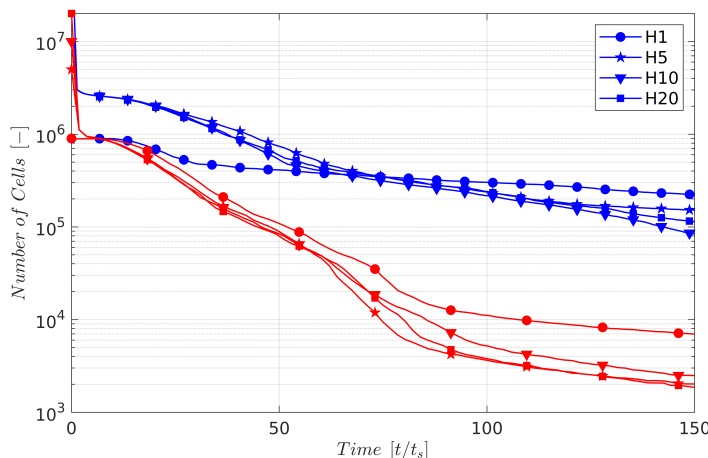

**Figure 13.** Total number of eDS type cells left in the simulation as a function of time for different $H\alpha$ cases. Lines are colored to represent the high-wind(UH, blue) and medium-wind (UM, red) scenarios. Note that the y-axis is logarithmic.

layer of differing depths. Each of these simulations begin with all the cells being of the eDS type which depending on their age, convert to neDS type cells during the course of the simulation. Given that the horizontal dimensions of the domain in each of our simulations is 1000 x 1000, the number of eDS type cells initially in a $H\alpha$ type simulation is $\alpha \times 10^6$ cells. The eDS type cells essentially contain the only snow mass available for transport, the rest being sintered and permanently deposited. Firstly, the number of eDS cells decreases as a function of wind speed as shown by the curves of different colors, blue lines being for the UH case, while the UM cases are represented by red lines. This implies that there is more permanent deposition as wind speed


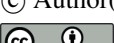

decreases - an admittedly intuitive result. For the same wind speed, comparing the results of snow layers of differing initial depths is rather counter-intuitive. The shallowest snow layer, H1 ( i.e. depth of 6.5 cm) seems to have the largest number of eDS type cells left in the latter stages of the simulation in the UH case and throughout the simulations in the UM case. This means that for medium to high wind speeds, shallow snow layers are more mechanically unstable and resist permanent deposition due to continual transport by wind. Furthermore, this difference between shallow and deep snow layers seems to increase with decreasing wind speed - at lower wind speeds, the shallow snow layers are comparatively more unstable than deeper snow layers. This result is extremely important for polar regions, particularly Antarctica where precipitation amounts are small and wind speeds are high. The analysis presented here highlights the fact that fresh snow fall in Antarctica is mechanically highly unstable.

## 5   Summary and Outlook

In the first section in this study, we performed a series of numerical experiments to investigate the morphodynamics of a solitary barchan dune. A range of barchan sizes were simulated under the action of three different wind regimes representing low, medium and high wind scenarios. Even without accounting for the effect of sintering, some new insights were gained since the scale of dunes simulated ($\mathcal{O}$10m), while relevant for snow bedforms, were an order-of-magnitude smaller than barchans found in sand ($\mathcal{O}$100m) , which are more well-studied. It was found that even small barchan dunes converge to Bagnold's model for barchan speed as a function of their height. However, this convergence is achieved for very long time averages ( between 18 and 36 hours depending on the dune size ). The instantaneous dune speeds have very large fluctuations and thus extracting any meaningful information from short time-series of dune speeds is extremely challenging. While long-time averaging is feasible in the framework of numerical experiments, it is highly unlikely that long-time series of barchan speeds can be collected in the field. We also show that the variance of dune speeds decrease with barchan size. Finally, the effect of barchan size on dune speed is found to be far more prominent than the effect of wind speed. Overall, the motion of dunes of dimensions ranging from (L,H) = (15.7 m, 0.4 m) to (60.5 m, 1.6 m) was simulated for three different wind speeds, namely, $U_{1m} = 20.5\mathrm{ms}^{-1}$, $12.5\mathrm{ms}^{-1}$ and $7.0\mathrm{ms}^{-1}$. The fastest dune had a velocity approaching 4 cm/min ( dune C14, UH case ) while the slowest dune had a velocity of 0.12 cm/min ( dune C44, UL case). The values of the dune speeds were quite similar to those reported in literature.

Accounting for the effect of sintering on morphodynamics of barchans previously in steady-state motion revealed three types of behaviour. Dunes smaller than a threshold size were found to continue their motion without any perceptible effect of sintering. On the other hand, dunes much larger than the threshold size were found to cease motion immediately upon the activation of the sintering effect. For barchans with sizes close to the threshold size, it was found that a part of the dune becomes immobilized and permanently deposited with the remainder of the dune maintaining its downwind motion. The threshold size is determined in terms of the maximum streamwise length (M.S.L) of any snow bedform (in this case a solitary dune). M.S.L is directly proportional to wind speed and to the sintering rate. We numerically found M.S.L to be equal to 21.5 m and 12.3 m



for the high ( UH, $U_{1m} = 20.5\mathrm{ms}^{-1}$ ) and the medium wind ( UM, $U_{1m} = 12.5\mathrm{ms}^{-1}$ ) cases. For the low wind cases, barchans of all sizes were immediately sintered "in-place" once sintering was activated and thus the M.S.L is less than 8.8 m.

In the following section (Sect. 4) we showed results of simulations of wind blowing over an initially flat surface of a snow layer of a finite depth. We considered snow layers with four different depths and two different wind speeds for our investigations (UH and the UM wind cases). The sintering process was activated from the beginning of the simulation unlike simulations in Sect. 3. This scenario is more realistic and can be considered as representing the situation of strong wind blowing after a relatively calm snowfall event. Each simulation was repeated by removing the sintering process, thus simulating a sand-like cohesionless material. This was done to clearly show the effect of sintering on snow bedforms. Qualitatively, we found that the initially flat and uniform snow layer reorganizes into a few mobile, erodible snow bedforms such as waves, transverse dunes and barchans. As time progresses, the dimensions of these bedforms as well as their number decreases due to sintering. New, non-erodible snow deposits are found throughout the domain. These deposits are much shallower than the mobile bedforms while having larger dimensions in the horizontal directions. We find that the concept of sintering imposing a maximum streamwise length for any mobile bedform, first elucidated in Sect. 3 remains valid in this scenario as well. Inspite of a large number of bedforms, each of which is changing its dimensions as well as the speed while at the same time interacting with each other, we find that the M.S.L in the domain is lower than or close to the limiting values found in Sect. 3 and described above.

Some additional valuable results are obtained as well. We find that whatever may be the depth of the fresh snow layer deposited, the amount of snow that remains erodible and thus available for snow transport remains the same in absolute numbers. We further find that shallow snow layers are more mechanically unstable as compared to deeper snow layers and this effect is more pronounced for lower wind speeds. This result is particularly interesting for regions with small precipitation amounts and moderate to strong winds. In such a location, snow may never be permanently deposited and be continuously blown !

Cellular automata based modelling for snow bedforms has being introduced in this study with the intention of modelling the effect of sintering on snow bedforms and ultimately deposition. There are indeed various aspects of this study that need to be developed and advanced further to cover a full range of scenarios that would occur in reality. Firstly, a more physically-based sintering model, suitable for the CA framework should be adopted. A simple extension in future work could be to implement sintering rate as a function of mean air temperature and overburden pressure. Secondly, it would be important to implement a transition type in the CA model to account for erosion by snow-grain impacts on sintered surfaces. This erosion mechanism is not currently taken into account and thus prevents us for simulating erosional features such as sastrugi. Future works will focus on these developments. Apart from these physical modelling improvements, numerical experiments could be performed over realistic topography underlying the snow layer, which would be especially interesting for snowfall deposition in complex terrain.

There are some caveats however to the CA-based modelling approach. The model parameters, consisting of different transition rates are free parameters that have been obtained essentially by trail-and-error. Upon performing a simulation, the time and length scales are established a-posteriori by relating unstable modes and fluxes to those provided by theoretical formulations. We were fortunate to have been aided by previous studies using this approach in the sand community who compared it with





field data in deserts. At a fundamental level however, there is a need to constrain each of these parameters independently to physically-based formulations. This could potentially be achieved by using large-eddy simulations (LES) of aeolian transport, the results of which are further translated into transition probabilities. In the present study, we found barchan speeds to be fairly close to the few measurements that exist in the literature. New field campaigns, such as the recently published study by

Kochanski (2018) , with a focus on surface morphology of snow surfaces, would be welcome in this regard for intercomparison and verification.

CA-based modelling could pave the way to estimate parameters such as momentum and scalar roughness lengths in a more robust manner. Recent studies have highlighted the influence of surface roughness, particularly due to sastrugi on momentum exchange between the air and the surface. One may also question the role of re-organization of snow and the formation of

snow bedforms on the albedo of the surface. Consider for example, the surfaces shown in Figs. 10d-f. An originally fully snow covered surface with a uniform depth of 5 cm has been transformed by the action of wind into vast regions where the underlying surface is exposed once again, whereas the erodible (and thus un-sintered) snow is localized in a few spots covering only a small portion of the overall surface area. How large would the differences in albedo of such surfaces be with and without accounting for wind-blow reorganization of snow ?

The ultimate goal of CA-based modelling efforts would be to couple surface morphodynamics with regional scale weather and climate models. CA-based modelling offers an extremely rapid, yet robust methodology that couples aeolian transport of material and evolving surface morphology while being tightly coupled with atmospheric flow that co-evolves with the topography. An additional advantage is that it can be coupled to atmospheric models in an offline manner given the difference in timescales involved, further easing it's adoption. As a future outlook, we feel that this methodology promises to be an

exciting new tool in snow-atmosphere interaction community.

*Acknowledgements.* We thank the authors of the RESCAL model for providing it freely with the right to use and modify. The study was funded by the 'Local Surface Mass Balance in East Antarctica' (LOSUMEA) project of the EPFL, the Swiss National Science Foundation (Grant n. 160667) and supported by the Swiss Super Computing Center (CSCS) which provided the computational resources (Project: s873). We thank Etienne Vignon, Alexis Berne and Fransiska Gerber for insightful discussions and Celine Labouesse for improving the manuscript.

The group's East Antarctica field campaigns are further supported by the National Institute of Polar Research, Japan and the Cryosphere Research Laboratory at Nagoya University, Japan (PI: Koichi Nishimura) and their help is gratefully acknowledged.





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
