# Peer review of "Understanding Snow Bedform Formation by Adding Sintering to a Cellular Automata Model"

_The Cryosphere, 2019_

## Short Comment (SC1) · 16 Apr 2019

Snow bedforms cover large parts of the polar regions of Earth. Sintering, the mechanism by which snow hardens over time, clearly influences bedform dynamics. The authors have studied the effects of sintering by adapting a cellular automaton model, ReSCAL, and have discovered that it limits the maximum streamwise length (MSL) of snow dunes. The authors also show a good awareness of the strengths and limitations of ReSCAL. The simulations they use are well designed, and the analysis is appropriate and thorough. Their simulation methods and results both agree well with my field observations, and I firmly believe that the processes they have modelled are real.

I've submitted this comment because I have studied snow dunes extensively in the

field, and want to ensure that the editor and future readers know that this study models reality well. I am pointing this out in a comment, rather than leaving it up to the authors, since most of the work was not published when the study was written, and I believe it is pertinent to point out the observations that strengthen their case.

I have also done some work with ReSCAL and have a few technical suggestions. I am not associated with the authors, but since this is not an 'extra' comment, not requested by the editor, I have tried not to suggest anything burdensome.

Yours, K. Kochanski

***Comparison to field data*** Thorough validation is clearly beyond the scope of this study, but I would like the authors and editors to see that these results capture important aspects of reality.

Fig. 6a, in which a dune is trailed by a patch of newly-sintered snow, looks a good deal like larger snow dunes I have observed in the field. See Figs. 2 and 6 of https://www.the-cryosphere.net/13/1267/2019/tc-13-1267-2019.html.

Most of the simulations used here represent the rare case in which snow dunes form well, without the influence of sintering, then begin to sinter abruptly. This happens occasionally in reality - I have one field example where dunes stop moving instantly when the temperature rises to 0, as in Fig. 6b (see https://www.youtube.com/watch?v=vFEwMPtO0pY) - but is probably rare.

Under constant temperature conditions, I expect that real snow dunes grow gradually but continuously like sand dunes grow. This has not been modelled or well-documented in the field, but if it's true, dunes will grow from nothing up to the MSL, when they will lose mass by sintering and grow no further. In this case, the MSL will be not only the maximum size of snow dunes but they usual size of well-developed dunes. This makes the MSL more important than the current text suggests.

***Specific comments on ReSCAL*** *Sintering mechanism* I was not clear, from Fig.

[Figure]

1 or the text, how your sintering mechanic is implemented. All existing ReSCAL transitions use doublets, or pairs, of cells (as shown in Fig. 1a) but your illustration of sintering uses single cells (Fig. 1d). Did you adapt ReSCAL to use single-cell transitions? If you used a doublet transition, this is important, because it would imply that your sintering mechanic works differently when snow is in contact with air or ground than with snow.

*Scaling ReSCAL* ReSCAL has three units: a length scale l0, a time scale t0, and a stress scale tau1. These units are the only way to convert ReSCAL results to real lengths, speeds, times, and forces. Getting them right is of critical importance and is the only way that this work can be applied to real snow or compared to field data. Moreover, since the values of tau1 and t0 vary between simulations, getting the tau1/t0 conversions right is the only way that these simulations can be compared to one another.

- The values in Table 1 and Table 2 must have uncertainties, and these uncertainties must be correctly propagated from l0 to tau1 and t0. In my experience with ReSCAL, reasonable values for l0 vary by a factor of 2-3 with snow grain size and density, and values for tau1 and t0 vary more. - The authors' value for l0 is about 50% too high. On p9 line 30, the authors calculate l0 by using a fairly low value for air density (1 kg/mˆ3, equivalent to freezing air at some ~2000m elevation) and the highest possible value for snow density (910 kg/mˆ3, the density of solid ice). This also shows up in the results; the authors have barchan snow dunes that are as much as 1.6 m high, when all the real dunes I've seen or read about are < 1m. - Since the value of l0 is used non-linearly to obtain the value of t0, the values for t0 will also be too high. - Adjusting l0 and t0 will also change all of the sizes/speeds listed in the text of the paper, and (in most cases I checked) will give better results when the authors compare their model to reality.

Correcting these values will allow the authors to get more accurate results, that better resemble reality and can be appropriately compared with field data, without needing to run more simulations or repeat their analysis.

*References* On p27 l5 you but cite raw data where I think you mean to cite the relevant published study (doi=10.1029/2018GL077616). If you do wish to cite a dataset you used, then (a) that's good practice, and (b) please include the full citation with the doi (10.5281/zenodo.1253725) so readers can find it too.

---

## Referee Comment (RC1) · Clement Narteau (Referee) · 24 Apr 2019

**tc-2019-45 Review**

**"Understanding Snow Bedform Formation by Adding Sintering to a Cellular Automata Model"**

by Varun Sharma et al.

This manuscript presents a numerical analysis of snow bedforms using a cellular automaton approach, which has been extensively used to study sand dune morphodynamics. The main originality of this work is to introduce a new mechanism to account for the cohesive properties of ice particles. This is achieved by a characteristic residence time above which the deposited snow particles can no longer be mobilized. Using this new model, the authors study in full details the emerging bedforms (shape, velocity) under different conditions of snow availability and wind strength. They convincingly show how consolidated barchans (low snow availability) and longitudinal structures (high snow availability) may develop. In addition to the modeling of these topographic features, they propose a threshold length-scale for the activity of the bedforms from which they develop.

The model is described in mathematical details and the first sections provide a comprehensive and accessible introduction to the feedback mechanisms that control the emergence of bedforms. The results are clearly presented and the structure of the manuscript allows a clear understanding of the different stages of the research.

I am convinced that this manuscript will be an inspiring and informative read for the cryosphere community and I highly recommend it for publication. I have just some comments below that, I hope, could help to give greater perspectives to this nice piece of work.

**Preliminary words**

Note that I am biased about the model because I have been developing this approach for almost 14 years. Nevertheless, I can appreciate the amount of work dedicated to this study and the quality of the research. The authors kept the model as simple as it is and only added a new level of complexity to explore some specific properties of snow bedforms. Thus, they have been able to built on prior knowledge to identify new patterns relevant to the evolution of icy landscapes. The precise evaluation of the characteristic length and time scale of the model makes it applicable to a wide range of conditions and provides a new tool for the analysis of practical issues. Therefore, this work will form the basis for further investigations of landforms in cold environments. In collaboration with Ghislain Picard, we wanted to study snow bedform with a similar approach. This manuscript demonstrate that it was a good idea.

I hope that the authors will help us to implement their model in the ReSCAL depository available online on a GNU General Public License. To do so, we can take advantage of the CELL\_TIME option in the most recent version of the code, which simply record the last motion of sedimentary/snow cells (see Fig. 6a of *Gao et al., 2015*). The CELL\_COLOR option may be also useful to decompose sedimentary/snow state into substates (see *Gao et al., 2016*).

**Main comments:**

 In order to inject the sintering mechanism in the model, I understand that the erosion rate of deposited snow cells writes

$$\Lambda_e(t) = \begin{cases} 0 & \text{for } \tau_s \leq \tau_1 \text{ or } t - t_{dep} \geq t_s, \\ \Lambda_0 \left( 1 - \frac{t - t_{dep}}{t_s} \right) \left( \frac{\tau_s - \tau_1}{\tau_2 - \tau_1} \right) & \text{for } \tau_1 \leq \tau_s \leq \tau_2 \text{ and } t - t_{dep} < t_s, \\ \Lambda_0 \left( 1 - \frac{t - t_{dep}}{t_s} \right) & \text{else.} \end{cases}$$

$$(1)$$

where  $t_s$  and  $t_{dep}$  are the sintering time and the deposition time of the corresponding cell, respectively. Please specify if it is the case or if you use a transition or a substate.

- To limit the aspect ratio of your bedforms, you consider that your elementary cell is a slab 5 times smaller in height than the length of its square base. In this case, you break the symmetry of the lattice-gas cellular automaton model and the momentum is not conserved during collisions. We have encountered the same problem in the past and, to solve it, we have increased the density of the square cells of the lattice-gas model in the horizontal direction (here for example by a factor 5). This option is still in the code but it has not been tested since 2010, before the first version of ReSCAL available online.

I admit that it will be difficult to solve this problem within the time frame of a review process and I am also convinced that it will not changed the results significantly. However, you must specify that there is a problem with the aspect ratio of your cell with respect to the air flow modeling.

 In dune dynamics, the relationship between dune height and speed writes

$$c = \frac{Q_{\text{crest}}}{(H+H_0)}$$

where  $H_0$  is a minimal dune height and

$$Q_{\text{crest}} = (1+\gamma)Q_{\text{sat}}.$$

In this expression,  $\gamma = \beta H/L$  is the speed-up factor which accounts for the increase in wind speed above a topographic obstacle. H/L is the dune aspect ratio and  $\beta$  a dimensionless coefficient that accounts for flow properties. Usually,  $\gamma$  is measured between 0.5 and 2 in nature. In the model,  $\gamma = 1.6$  (*Gao et al.*, 2015a).

Then, instead of Eq.6 of the manuscript you should find the best fit using a relation of the form

$$c^* = \frac{c}{Q} = \frac{a}{H + H_0}.$$

I predict that it will fit the data rather well with  $a \approx 2.6$  and  $H_0 \rightarrow 0$ . Most importantly, the *a* and  $H_0$ -values will have a physical meaning, the speed-up and a minimum dune size respectively.

- The characteristic time scale  $t_{dune}$  of a dune should scale with  $H^2/Q_{sat}$ . It can be described as the dune turnover time or as the time it takes for a dune to loss the memory of its shape. This characteristic time should be compared to the  $t_{\rm s}$ -value. I guess that for  $t_{\rm dune} < t_{\rm s}$ , dunes will remain mobile. For  $t_{\rm dune} > t_{\rm s}$ , dunes will sinter. Then, it could be informative to test if the maximum streamwise length discussed in the manuscript scales as  $\sqrt{t_s Q_{\rm sat}}$ .

 $\frac{\text{Minor comments:}}{\text{Line } n \text{Pm is for Line } n \text{ of Page } m.$

- Lines 3P1, 26P4, 2P9, 25P26 and caption Fig. 1: Specify that you implement a "sintering mechanism" and not a "sintering model".
- Line 6P9: Define  $t_{\rm s}$ .
- Line 2P3: On the basis on the equation of conservation of mass  $\partial Q/\partial x = -\partial h/\partial t$ , erosion and deposition should be associated with increasing or decreasing transport. In a second time, you can specify that transport is positively correlated to the wind shear stress.
- Section 2: Rozier and Narteau (2014) should be cited as they introduce the ReSCAL model in a more general research perspective, in particular for multidisciplinary studies in landscape dynamics. For information, Narteau et al. (2001) have used a preliminary version of ReSCAL to study dissolution/crystallization mechanisms, which may be of wide interested in icy landscapes where melting/freezing processes are likely to play a crucial role.
- Line 26P5: check spaces after and before parentheses. Similarly, you can remove the math mode for subscripts and upperscripts, for example  $Q_{\text{sat}}$  instead of  $Q_{\text{sat}}$ .
- Line 27P9: In linear stability analysis, we measure the exponential growth of the amplitude with respect to time  $(d \ln(A)/dt)$ .
- Snow falls are easy to implement using INPUT cells.
- Line32p26: using a simplified version of the ReSCAL dune model, Zhang et al. (2014) have compared the rate parameter to physically-based formulations,

**Bibliography**

- Gao X., C. Narteau, O. Rozier, *Development and steady states of transverse dunes: A numerical analysis of dune pattern coarsening and giant dunes, Journal of Geophysical Research Earth Surface,* **120**, 2200–2219 (2015).
- Gao X., C. Narteau, O. Rozier, S. Courrech du Pont, Phase diagrams of dune shape and orientation depending on sand availability, Scientific Reports, 5, 14677 (2015a).
- Gao X., C. Narteau, O. Rozier, Controls on and effects of armoring and vertical sorting in aeolian dune fields: A numerical simulation study, Geophysical Research Letters, 43, 2614-2622, (2016).
- Narteau C., J.-L. Le Mouël, J.P. Poirier, E. Sepulveda and M. Shnirman, On a small scale roughness of the core-mantle boundary, Earth and Planetary Science Letters, **191**, 49–60 (2001).
- Rozier O. and C. Narteau, A real space cellular automaton laboratory. **39**, 98–109 (2014).
- Gao X., Zhang D., Rozier O., Narteau C., Transport capacity and saturation mechanism in a real-space cellular automaton dune model, Advances in Geosciences, **37**, 40–49 (2014).

---

## Referee Comment (RC2) · Anonymous Referee #2 · 27 May 2019

Review of "Understanding Snow Bedform Formation by Adding Sintering to a cellular Automata Model" by V. Sharm, L. Braud, and M. Lehning.

This article presents the use of two coupled cellular automata models to simulate the snow bedform formation under different wind conditions and different initial snowpack depths. The authors focused on the impact of sintering (through a simple sintering model) on the shape and characteristics of snow bedforms. It was showed that sintering has an important impact on snow bedforms for larger snow dunes and little impacted the motion of smaller dunes. This article does not present any validation of the model but shows realistic results.
I find this paper well written and well structured. Despite the lack of validation of the model, I believe this paper is well suited for publication in The Cryosphere.

I am not familiar with the cellular automata model but the authors made a good effort in describing all of the components of the two CA models and all the simulations done.
I only have minor comments that I hope will make some sections of this interesting paper a bit clearer.

P.1 L.20: "a year"

P.2 L.7-9: I don't find this sentence very clear.

P.3 L.5: "is the fact that"

P.3 L.7: "time-scales of snow transport are much shorter"

P.5 L.15: I understand the reason to use a finer grid in the vertical resolution than in the horizontal resolution. Could the other explain the choice of a ratio of 5 or talk about the potential impact of this ratio on model results?

P.5 L.19: "known"

Please correct the added spaces before and after parentheses.

P.5 L.26: you are introducing the transition-rate parameters here and then taking again about them again in P.7 L.33 (and Table 1, P.8). Please introduce the notation $\Lambda$ for the transition-rate parameter in the last paragraph of page 5.

P.5 L.10 and P.7 L.3: I would change the names of the subsections to make them more descriptive. For instance, Section 2.1 could be named "Description of the CA model for snow transport" and the Section 2.2 could be "Description of the LGCA model for snow surface evolution".

Section 2.1: For a reader not familiar to CA modelling, it is difficult to understand how the transitions happen for the doublets. Fig. 1a is not helping to explain these transitions and the meaning of the variables $\Lambda$, $b$, and $\delta$ is not explained until later in the text.

P.7. L.14: Please specify that the direction and number of particles are represented by the arrows in Fig 1b.

P.7 L.14: "collision rules". What are these rules? At least a citation describing these rules would be needed.

P.9 L.11: "Fig. 1d"

Fig. 1d needs a better description. What do the boxes mean? In addition, the variable $t_s$ should be introduced in P.9, first paragraph, when introducing the 24h sintering time.

In all graphs expressing time, the unit showed by the authors is $[t/t_s]$. I believe it should be "[-]" (or "[s/s]") and the label in the x-axis should state "Normalized time ($t/t_s$) [-]".

Similar comment about the units as above for the variables in Fig. 2.

P.10 L.8: What is the meaning of the threshold velocity? The velocity at which erosion starts?

Section 3.1: Please make it clear in the first paragraph if sintering is turned on or not.

End of P. 13 + Fig. 4a + P. 14:  I believe Fig. 4 shows the speed of the dune vs. time for different dune heights (see caption of Fig. 4a). In the text and in the legend of the figure, it is not clear that this graph presents results for different dune heights. Indeed, the legend of the figure shows the length of the dune "L" and not "H". In the text, the authors talk about "dune length" (P.13 L.13) and "dune size" (P. 14 L.7).

P.16 L.11: "the dune is deposited as a non-erodible layer"

P.17 L.7: "this behaviour becomes clear"

P.17 L.25: "These simulations allow us to identify"

Fig. 7: It is hard to see the legend of the color bars of the left graphs.

For all the simulations, I could not find any information on the initial conditions for the wind speed. Is it initially 0 m/s and then the left boundary condition is where the wind speed is set or is the wind speed initially set to the chosen value over the whole area?

There is no information about the snow properties (e.g. snow density and grain size) used for the simulations. How do they impact the snow bedform formation?

I do not understand how dunes form in the simulations presented in Section 4. If the snow surface is initially flat and the wind speed is constant, how do the first snow dunes form? It seems to me that some sort of heterogeneity would be needed for the first dunes to form and then propagate.

---

## Author Comment (AC1) · 5 Aug 2019

**Response to reviewers:**
**Understanding snow bedform formation by adding sintering to a cellular automata model**

Varun Sharma, Louise Braud and Michael Lehning

August 4, 2019

**Response to Reviewer # 1**

**Opening Remarks:**

We were extremely pleased to have Prof. Clement Narteau as a reviewer for our manuscript. Considering that he is the principal developer of the RESCAL model and has played a major role in expanding cellular automata approaches in the sand dune community, his positive remarks on our work are heartening. Indeed, we would be very happy to have our implementation of sintering be a part of the next release of RESCAL as Prof. Narteau proposes.

We thank Prof. Narteau for his comments and advice and have incorporated most of them into the updated manuscript. What follows is a point-by-point response to the questions and comments made in the review.
* * *
**A: Main Comments**

- **A.1 : In order to inject the sintering mechanism in the model, I understand that the erosion rate of deposited snow cells writes**

$$\Lambda_e(t) = \begin{cases} 0 & \text{for } \tau_s \leq \tau_1 \text{ or } t - t_{dep} \geq t_s \\ \Lambda_0 \left(1 - \frac{t - t_{dep}}{t_s}\right) \left(\frac{\tau_s - \tau_1}{\tau_2 - \tau_1}\right) & \text{for } \tau_1 \leq \tau_s \leq \tau_2 \text{ or } t - t_{dep} \leq t_s \\ \Lambda_0 \left(1 - \frac{t - t_{dep}}{t_s}\right) & \text{otherwise} \end{cases} \tag{1}$$

  **where $t_s$ and $t_{dep}$ are the sintering time and the deposition time of the corresponding cell, respectively. Please specify if it is the case of if you use a transition or a substrate.**

  **Response A.1:** We have used the transition-blocking technique that is a part of the RESCAL model already. The CELLTIME variable that is available for each deposited cell is used to track the time of deposition and the age of the cell. Your equation is thus, exactly what we have implemented.

  We expanded the description of the sintering mechanism into a new sub-section and added a version of the above equation to it. We did so by introducing a new variable called *Erodibility factor*, $f_E$. Correspondingly, Fig. 1d's y-axis has also been updated.

- **A.2 : To limit the aspect ratio of your bedforms, you consider that your elementary cell is a slab 5 times smaller in height than the length of its square base. In this case, you break the symmetry of the lattice-gas cellular automaton model and the momentum is not conserved during collisions. We have encountered the same problem in the past and, to solve it, we have increased the density of the square cells of the lattice- gas model in the horizontal direction (here for example by a factor 5). This option is still in the code but it has not been tested since 2010, before the first version of ReSCAL available online. I admit that it will be difficult to solve this problem within the time frame of a review process and I am also convinced that it will not changed the results significantly. However, you must specify that there is a problem with the aspect ratio of your cell with respect to the air flow modeling.**

**Response A.2:** We mention in the text that the lattice-gas model is performed on a grid with size equal to the smaller of the two dimensions. Infact, we did exactly as you have done previously and mentioned in your comment. We chose the necessary parameters in the lgca.h file of the source code.
* * *
- **A.3 : In dune dynamics, the relationship between dune height and speed writes as**

$$c = \frac{Q_{crest}}{(H + H_0)} \,, \tag{2}$$

where $H_0$ **is a minimal dune height and**

$$Q_{crest} = (1 + \gamma)\, Q_{sat} \,. \tag{3}$$

In this expression, $\gamma = \beta H/L$ is the speed-up factor which accounts for the increase in wind speed above a topographic obstacle. H/L is the dune aspect ratio and $\beta$ a dimensionless coefficient that accounts for flow properties. Usually, $\gamma$ is measured between 0.5 and 2 in nature. In the model, $\gamma = 1.6$ (Gao et al. 2015a). Then, instead of Eq.6 of the manuscript you should find the best fit using a relation of the form,

$$c^* = \frac{c}{Q} = \frac{a}{H + H_0} \,. \tag{4}$$

I predict that it will fit the data rather well with a $\approx$ 2.6 and $H_0 \to 0$. Most importantly, the $a$ and $H_0$ values will have a physical meaning, the speed-up and a minimum dune size respectively.

**Response A.3:** This is indeed an interesting question. We infact began with the expression you propose in your comment (and have earlier published on). However, we realized that the two parameter hyperbolic equation was only able to fit the data for small dunes, that is, for small H values. Please find attached, an updated figure with an additional fit as you propose (red line).

What is even more interesting is the fact that the values of $a_{Narteau}$ and $H_{0,Narteau}$ are close to what you predict, 2.694 and 0.02101 respectively. However, we are unable to explain the lack of good fit for larger H values. There are multiple possible reasons for this divergence. However, We feel that this may require more exploration that we leave for the future.

[Figure]

Figure 1: See the additional fit line.
* * *
- **A.4 : The characteristic time scale $t_{dune}$ of a dune should scale with $H^2/Q_{sat}$. It can be described as the dune turnover time or as the time it takes for a dune to loose the memory of its shape. This characteristic time should be compared to the $t_s$-value. I guess that for $t_{dune} < t_s$, dunes will remain mobile. For $t_{dune} > t_s$, dunes will sinter. Then it could be informative to test if the maximum streamwise length discussed in the manuscript scales as $\sqrt{t_s\,Q_{sat}}$.**

**Response A.4:** We would like to discuss this topic in a future publication, particularly with real-world field data to test. We were thinking on similar lines but decided to focus on the concept of the maximum streamwise length in this manuscript. The current manuscript was intended to introduce the CA approach to the snow pack community. With the RESCAL model, a lot of topics are left to be explored in the context of snow!
* * *
**B: Minor comments**

- **B.1 : Lines 3P1, 26P4, 2P9, 25P26 and caption Fig. 1: Specify that you implement a "sintering mechanism" and not a "sintering model".**

**Response B.1:** We have replaced "sintering model" with the "sintering mechanism" in the text of the updated manuscript.
* * *
- **B.2 : Line 6P9: Define $t_s$ .**

**Response B.2:** As mentioned above for point A.1, in the new sub-section of the sintering mechanism, we additionally define the sintering time scale $t_s$.

- **B.3 : Line 2P3: On the basis on the equation of conservation of mass $\partial Q/\partial x = \partial h/\partial t$, erosion and deposition should be associated with increasing or decreasing transport. In a second time, you can specify that transport is positively correlated to the wind shear stress.**

  **Response B.3:** This concept has been described in the updated manuscript.
* * *
- **B.4 : Section 2: Rozier and Narteau (2014) should be cited as they introduce the ReSCAL model in a more general research perspective, in particular for multidisciplinary studies in landscape dynamics. For information, Narteau et al. (2001) have used a preliminary version of ReSCAL to study dissolution/crystallization mechanisms, which may be of wide interested in icy landscapes where melting/freezing processes are likely to play a crucial role.**

  **Response B.4:** We have now cited Rozier and Narteau (2014) at the beginning of Section 2.
* * *
- **B.5 : Line 26P5: check spaces after and before parentheses. Similarly, you can remove the math mode for subscripts and upperscripts, for example $Q_{\mathrm{sat}}$ instead of $Q_{sat}$.**

  **Response B.5:** Thank you for pointing this editing mistake. We have removed the unnecessary spaces before and after parentheses along with updating all the subscripts and upperscripts to be in normal mode.
* * *
- **B.6 : Line 27P9: In linear stability analysis, we measure the exponential growth of the amplitude with respect to time (d ln(A)/dt).**

  **Response B.6:** Thank you for pointing out this mistake. We have corrected this both in the text as well as the y-axis in Figure 2a.
* * *
- **B.7 : Snow falls are easy to implement using INPUT cells.**

  **Response B.7:** That is indeed true. We have already made use of it in our ongoing work!
* * *
- *B.8 :Line32p26: using a simplified version of the ReSCAL dune model, Zhang et al. (2014) have compared the rate parameter to physically- based formulations,*

**Response B.8:** The necessary line and citation as been added in the updated manuscript.

---

## Author Comment (AC2) · 5 Aug 2019

**Response to reviewers:**
**Understanding snow bedform formation by adding sintering to a cellular automata model**

Varun Sharma, Louise Braud and Michael Lehning

August 5, 2019

**Response to Kelly Kochanski's Short Comment**

We thank Kelly Kochanski for taking the time to go through our manuscript and make important comments on the both the technical and scientific side of our work.

Kelly Kochanski has extensive field experience in studying snow bedforms in Colorado and her previous published work in The Cryosphere (https://www.the-cryosphere.net/13/1267/2019/) is an impressive collection of data and analysis. This makes her insight and critique of our work even more pertinent. Furthermore, she too is developing a snow specific version of ReSCAL (https://github.com/kellykochanski/rescal-snow) which is quite an exciting development.

We shall attempt to answer and clarify some of the issues she has touched upon below but her initial vote of confidence gives us further hope for continuing to explore the ReSCAL tool.

- It is quite heartening to note that you have observed sintered patches of snow trailing a mobile barchan dune. We must look into your datasets in more detail in the coming weeks.

- The main purpose of the cone experiments was to identify the M.S.L and not really to reflect reality. Our main thesis is that without sintering and with adequate supply of snow, snow barchans can grow like sand barchans. We turned on sintering abruptly to see whether a barchan of a particular size can even remain mobile for a given wind speed. A barchan that cannot remain mobile can in reality never grow to reach that size. We shall come back to this point further on.

- We realized that our sintering implementation is not clear in the submitted manuscript and we have made an attempt to make it clearer in the revised version. Since you are also a ReSCAL developer, I feel comfortable in being a bit more specific in responding to your question. The GR cells in ReSCAL come with an additional variable called CELL_TIME attached (see cells.h). This capability needs to be activated in defs.h. Whenever a cell transitions to GR, the CELL_TIME variable is updated to the current time-step value. We used this variable to keep a track of the "age" of the cell, i.e., how long as the cell been in place. You are perhaps using this variable in a similar way as well for your own sintering implementation.

  Next, we started blocking transitions of these GR cells in a stochastic fashion, similar to how the erosion transition is implemented in ReSCAL. For example, the probability of an erosion transition to occur for a particular GR cell is 0 if the age of the cell is greater than a chosen time, say 24 hours and the probability is 1 if the cell is subject to sufficient stress at the surface and it is freshly deposited, i.e., the age is 0 hrs. For cells with ages in between, we use a linearly decreasing probability.

- You are indeed correct about the sensitivity of the model with respect to the length and time scales chosen. It was one of the biggest challenges for us, particularly since we wanted to impose "realistic" sintering timescales. It was imperative for us to get a good handle on length and time scales. The air density we chose is indeed low and we could present a more extensive sensitivity analysis in the future. With regards to ice density, we are bit more confident about this number. Note that the threshold friction velocity and the flux formulations that are used to derive the length and time scales are based on density of drifting and blowing snow particles and not the bulk density of the granular bed, as far as my understand of Bagnold's book goes, or for that matter, drifting and blowing snow as described by Nishimura and Hunt, JFM 2000. The density of snow particles in aeolian transport should not be that far from that of solid ice. I am more concerned about the particle diameter which is likely to be larger than that in reality.

  With regards to our 1.6 meters high barchans - these were conical pile experiments without sintering activated. In a hypothetical scenario, with a huge conical pile of snow and sufficient wind speed, there is no reason why there should not be a huge barchan that scales with the size of the initial conical pile - As long as there is no sintering. As soon as sintering was activated, the large barchans immediately ceased moving - another way of showing that in fact, such barchans can never even be formed.

  A more realistic picture is perhaps found in Fig. 9. Here we begin a flat, 1.3 meter layer of snow that evolves into a complex dune field under the action of a (very!) strong wind. The resultant dune field, after 50 days, is shown in Fig. 9b. While we have not presented numbers about the geometry of the dunes formed, most of the dunes are less than 1 m high. The largest dune is approaching 1.1 meters. The situation simulated however is unlikely to ever occur in reality - 20 m/s continuous wind at 1 m above the surface for 50 days - but data from Antarctica always manages to surprise us about how extreme it can get - may be it does in reality - but surely there is no data for snow bedforms for such a scenario.

- Many apologies for the sloppiness in citing your work. It has been corrected.

---

## Author Comment (AC3) · 5 Aug 2019

The comment was uploaded in the form of a supplement:
https://www.the-cryosphere-discuss.net/tc-2019-45/tc-2019-45-AC3-supplement.pdf

---

## Author Comment (AC4) · 5 Aug 2019

**Response to reviewers:**
**Understanding snow bedform formation by adding sintering to a cellular automata model**

Varun Sharma, Louise Braud and Michael Lehning

August 5, 2019

**A note to all reviewers**

**Response to Reviewer # 2**

**Opening Remarks:**

We thank Reviewer# 2 for his/her mostly positive comments and for finding the manuscript suitable for publication in The Cryosphere.

Below, we address the points of criticism raised by Reviewer# 2.
* * *
**A: Concerns in the main text**

- **A1 : " a year "**

  **Response A.1:** Corrected in the updated manuscript.
* * *
- **A.2 : P2 L7-9: I don't find this sentence very clear.**

  **Response A.2:** We updated these lines in the revised manuscript as follows.
  *The importance of snow bedforms primarily stems from the fact that their presence results in an undulating surface which affects basic exchange parameters that dictate transfer of mass, energy and momentum between the surface and the atmosphere, namely the roughness lengths for specific humidity, sensible heat and velocity.*
* * *
- **A.3 : P.3 L.5: "is the fact that"**

  **Response A.3:** Corrected in the updated manuscript.

- **A.4 : P.3 L.7: "time-scales of snow transport are much shorter "**

  **Response A.4:** Corrected in the updated manuscript.
* * *
- **A.5 : P.5 L.15: I understand the reason to use a finer grid in the vertical resolution than in the horizontal resolution. Could the other explain the choice of a ratio of 5 or talk about the potential impact of this ratio on model results?**

  **Response A.5:** The ratio of 5 is chosen as a balance between excessive computational expense, that increases with the resolution ratio and the accurate representation of the surface. Tests with a higher resolution ratio showed that the results were not significantly different. This is now explained in the revised manuscript.
* * *
- **A.6 : P.5 L.19: "known". Please correct the added spaces before and after parentheses.**

  **Response A.6 :** Corrected in the updated manuscript. The spaces before and after the parentheses have been removed. Thank you for pointing our this discrepancy.
* * *
- **A.7 : P.5 L.26: you are introducing the transition-rate parameters here and then taking again about them again in P.7 L.33 (and Table 1, P.8). Please introduce the notation $\Lambda$ for the transition-rate parameter in the last paragraph of page 5.**

  **Response A.7:** We now introduce the notation of $\Lambda$ earlier. Additionally, we point to Table 1 in the same paragraph unlike much later as in the original manuscript.
* * *
- **A.8 : P.5 L.10 and P.7 L.3: I would change the names of the subsections to make them more descriptive. For instance, Section 2.1 could be named "Description of the CA model for snow transport" and the Section 2.2 could be "Description of the LGCA model for snow surface evolution".**

  **Response A.8:** We agree with your comment and have changed the section titles appropriately.
* * *
- **A.9 : Section 2.1: For a reader not familiar to CA modelling, it is difficult to understand how the transitions happen for the doublets. Fig. 1a is not helping to explain these transitions and the meaning of the variables $\Lambda$, $b$, and $\delta$ is not explained until later in the text.**

  **Response A.9:** On this point, considering that the CA model has been described in many publications in the past, our intention was to only give a general idea of the CA modelling technique while pointing to the relevant literature. To explain the model in detail would require the manuscript to extended by many more pages, while simply repeating or paraphrasing earlier works. We instead chose to focus on the results of such a model and illustrate its capabilites both qualitatively (with attached videos for example) as well as quantitatively.

  ──────────────────────

- **A.10 : P.7. L.14: Please specify that the direction and number of particles are represented by the arrows in Fig 1b.**

  **Response A.10:** The description of Figure 1b has been updated in the revised manuscript. In particular the significance of the green arrows is better explained. Furthermore, a citation is added for the N-body collision rules.

  ──────────────────────

- **A.11 : P.7 L.14: "collision rules". What are these rules? At least a citation describing these rules would be needed.**

  **Response A.11:** See response to A.10 above.

  ──────────────────────

- **A.12 : P.9 L.11: "Fig. 1d". Fig. 1d needs a better description. What do the boxes mean? In addition, the variable $t_s$ should be introduced in P.9, first paragraph, when introducing the 24h sintering time.**

  **Response A.12:** In response to your comment, which is also shared by the other reviewer, we have expanded the description of the sintering mechanism by introducing a new subsection. The model description is more detailed now, along with a better description of Fig. 1d as well as defining $t_s$ which indeed an oversight.

  ──────────────────────

- **A.13 : In all graphs expressing time, the unit showed by the authors is [$t/t_s$]. I believe it should be "[-]" (or "[s/s]") and the label in the x-axis should state "Normalized time ($t/t_s$) [-]". Similar comment about the units as above for the variables in Fig. 2.**

  **Response A.13:** We have modified the x-axis labels in almost all plots to be more clear about the normalized time or age variables. We thank you for pointing out the inconsistencies in this regard in the original manuscript.

- **A.14 : P.10 L.8: What is the meaning of the threshold velocity? The velocity at which erosion starts?**

  **Response A.14:** $u_c$ is the threshold friction velocity for aeolian transport. These lines have now been added to the revised manuscript.
* * *
- **A.15 : Section 3.1: Please make it clear in the first paragraph if sintering is turned on or not.**

  **Response A.15:** We have not described this clearly in the revised manuscript.
* * *
- **A.16 : End of P. 13 + Fig. 4a + P. 14: I believe Fig. 4 shows the speed of the dune vs. time for different dune heights (see caption of Fig. 4a). In the text and in the legend of the figure, it is not clear that this graph presents results for different dune heights. Indeed, the legend of the figure shows the length of the dune "L" and not "H". In the text, the authors talk about "dune length" (P.13 L.13) and "dune size" (P. 14 L.7).**

  **Response A.16:** Thank you for pointing out these discrepancies. We stick to 'dune length' in the legend as well as the text.
* * *
- **A.17 : P.16 L.11: "the dune is deposited as a non-erodible layer"**

  **Response A.17:** Corrected in the updated manuscript.
* * *
- **A.18 : P.17 L.7: "this behaviour becomes clear"**

  **Response A.18:** Corrected in the updated manuscript.
* * *
- **A.19 : P.17 L.25: "These simulations allow us to identify"**

  **Response A.19:** Corrected in the updated manuscript.
* * *
- **A.20 : Fig. 7: It is hard to see the legend of the color bars of the left graphs.**

  **Response A.20:** We have updated with figure with larger color bars that are indeed more readable.

  ———————————

- **A.21 : For all the simulations, I could not find any information on the initial conditions for the wind speed. Is it initially 0 m/s and then the left boundary condition is where the wind speed is set or is the wind speed initially set to the chosen value over the whole area?**

  **Response A.21:** The initial velocity is indeed equal to 0 m/s. The LGCA model is run initially without any aeolian transport for the wind speed to accelerate and reach equilibrium with the initial surface morphology. Only once the wind speeds are statistically homogenous is the surface morphology allowed to evolve.

  ———————————

- **A.22 : There is no information about the snow properties (e.g. snow density and grain size) used for the simulations. How do they impact the snow bedform formation?**

  **Response A.22:** The snow properties used in all the simulations were described on Line 31 of Page 9 in the original manuscript.

  While it is likely that these parameters would have some influence over the surface features, these numbers are by themselves highly constrained. Note the density of the solid phase refers to density of ice grain rather than bulk density of the snowpack, which, admittedly is quite variable.

  ———————————

- **A.23 : I do not understand how dunes form in the simulations presented in Section 4. If the snow surface is initially flat and the wind speed is constant, how do the first snow dunes form? It seems to me that some sort of heterogeneity would be needed for the first dunes to form and then propagate.**

  **Response A.23:** In a modelling or even theoretical framework, it has been shown that a perfectly flat granular bed can be transformed into an undulating surface with ripples and waves. The initial heterogeneity need not come from the surface but from the overlying turbulent fluid that is indeed heterogeneous. This heterogeneity can be evidenced by the fact the instantaneous surface shear stress varies significantly in both time and space over a perfectly flat, smooth surface even if mean stress converges to a fixed value. Aeloian transport and particularly erosion is controlled not by averaged but instantaneous shear stresses at the surface. As soon as aeloian transport commences, bedform structures can develop. After a certain time, the structures grow to be large enough to significantly perturb the air flow and the feedback between the morphodynamics of the surface and the overlying flow become the dominant mechanism.

In reality, indeed the granular bed shall consist of heterogeneities - even a surface which is 'flat' at large scales will have heterogeneities at least of the diameter/pore scale by definition. But these heterogeneties are still much smaller than the length scale of even the smalled ripples that form.